# Dynamics of allosteric regulation of the phospholipase C-γ isozymes upon recruitment to membranes

Edhriz Siraliev-Perez[1], Jordan TB Stariha[2], Reece M Hoffmann[2], Brenda RS Temple[1], Qisheng Zhang[1,3,4], Nicole Hajicek[5], Meredith L Jenkins[2], John E Burke[2,6]*, John Sondek[1,4,5]*

[1]Department of Biochemistry and Biophysics, School of Medicine, University of North Carolina at Chapel Hill, Chapel Hill, United States; [2]Department of Biochemistry and Microbiology, University of Victoria, Victoria, Canada; [3]Division of Chemical Biology and Medicinal Chemistry, School of Pharmacy, University of North Carolina at Chapel Hill, Chapel Hill, United States; [4]Lineberger Comprehensive Cancer Center, School of Pharmacy, University of North Carolina at Chapel Hill, Chapel Hill, United States; [5]Department of Pharmacology, School of Medicine, University of North Carolina at Chapel Hill, Chapel Hill, United States; [6]Department of Biochemistry and Molecular Biology, The University of British Columbia, Vancouver, Canada

**Abstract** Numerous receptor tyrosine kinases and immune receptors activate phospholipase C-γ (PLC-γ) isozymes at membranes to control diverse cellular processes including phagocytosis, migration, proliferation, and differentiation. The molecular details of this process are not well understood. Using hydrogen-deuterium exchange mass spectrometry, we show that PLC-γ1 is relatively inert to lipid vesicles that contain its substrate, phosphatidylinositol 4,5-bisphosphate (PIP$_2$), unless first bound to the kinase domain of the fibroblast growth factor receptor (FGFR1). Exchange occurs throughout PLC-γ1 and is exaggerated in PLC-γ1 containing an oncogenic substitution (D1165H) that allosterically activates the lipase. These data support a model whereby initial complex formation shifts the conformational equilibrium of PLC-γ1 to favor activation. This receptor-induced priming of PLC-γ1 also explains the capacity of a kinase-inactive fragment of FGFR1 to modestly enhance the lipase activity of PLC-γ1 operating on lipid vesicles but not a soluble analog of PIP$_2$ and highlights potential cooperativity between receptor engagement and membrane proximity. Priming is expected to be greatly enhanced for receptors embedded in membranes and nearly universal for the myriad of receptors and co-receptors that bind the PLC-γ isozymes.

*For correspondence: jeburke@uvic.ca (JEB); sondek@med.unc.edu (JS)

## Editor's evaluation

This work provides insight into how phospholipase C-γ1 (PLC-γ1) becomes activated upon binding to phosphorylated receptor tyrosine kinase, with an analysis of PLC-γ1 bound to the soluble kinase domain of *FGFR1* (FGFR1K) and/or liposomes containing PIP$_2$. The most interesting finding is that regions of the protein far from the FGFR1K binding site increase in exchange upon binding. This is new information for a large protein that is arguably difficult to study, but it conforms to what has been observed in many other autoinhibited systems with similar SH2 and SH3 domains such as kinases. The results will be of interest to structural biologists and cell biologists with interest in the mechanisms leading to the regulation of phospholipase C activity on membranes.

## Introduction

The 13 mammalian phospholipase C (PLC) isozymes hydrolyze the minor membrane phospholipid phosphatidylinositol 4,5-bisphosphate (PIP$_2$) to create the second messengers inositol 1,4,5-trisphosphate (IP$_3$) and diacylglycerol (DAG) (*Gresset et al., 2012*; *Kadamur and Ross, 2013*). IP$_3$ diffuses throughout the cytosol and binds to IP$_3$-gated receptors in the endoplasmic reticulum, resulting in an increase in the concentration of intracellular calcium ions. In contrast, DAG remains embedded in the inner leaflet of the plasma membrane, where it recruits and activates multiple signaling proteins containing C1 domains including the conventional isoforms of protein kinase C (PKC). In addition, DAG is a precursor of phosphatidic acid, itself an important signaling lipid. PLC-dependent depletion of PIP$_2$ levels in the plasma membrane also regulates the activity of numerous ion channels and signaling proteins. Accordingly, the PLCs orchestrate fluctuations in the abundance of both phospholipids and second messengers to control important biological processes including proliferation (*Noh et al., 1995*) and cell migration (*Asokan et al., 2014*).

The PLC-γ isozymes, PLC-γ1 and PLC-γ2, are activated downstream of myriad cell surface receptors and regulate signaling cascades that modulate multiple aspects of embryonic development, directed cell migration, and the immune response. Mounting evidence also indicates that the PLC-γ isozymes are drivers of human disease including cancer and immune disorders (*Koss et al., 2014*). For example, PLC-γ1 is overexpressed and presumably activated in breast cancer (*Arteaga et al., 1991*). In addition, genome-wide sequencing studies have identified somatic gain-of-function mutations in PLC-γ1 and PLC-γ2 in a variety of immunoproliferative malignancies. In one cohort of patients with adult T-cell leukemia/lymphoma, PLC-γ1 is the most frequently mutated gene with almost 40% of patients harboring at least one substitution in the isozyme (*Kataoka et al., 2015*). Moreover, mutations in PLC-γ2 arise in approximately 30% of patients with relapsed chronic lymphocytic leukemia after treatment with ibrutinib, a covalent inhibitor of Bruton's tyrosine kinase (*Woyach et al., 2014*). Recently, a putative gain-of-function variant was identified in PLC-γ2, Pro522Arg, that was associated with a decreased risk of late-onset Alzheimer's disease (*Sims et al., 2017*). This variant also seems to slow cognitive decline in patients with mild cognitive impairment (*Kleineidam et al., 2020*), suggesting that in some contexts, elevated PLC-γ activity is beneficial.

Among the PLCs, PLC-γ1 and PLC-γ2 are distinct in that they are the only isozymes directly activated by tyrosine phosphorylation. It is widely accepted that phosphorylation of Tyr783 in PLC-γ1 and the equivalent site in PLC-γ2, Tyr759, is required for their regulated activation (*Kim et al., 1991*; *Kim et al., 1990*; *Watanabe et al., 2001*). Phosphorylation and consequently, activation of the PLC-γ isozymes are mediated by two major classes of tyrosine kinases. Multiple receptor tyrosine kinases (RTKs) including receptors for growth factors such as fibroblast growth factor (*Burgess et al., 1990*), as well as Trk receptors (*Vetter et al., 1991*) activate the PLC-γ isozymes. PLC-γ1 and PLC-γ2 are also phosphorylated by soluble tyrosine kinases associated with immune receptors, including members of the Src, Syk, and Tec families (*Law et al., 1996*; *Nakanishi et al., 1993*; *Schaeffer et al., 1999*).

PLC-γ1 and PLC-γ2 contain a distinctive array of regulatory domains that controls their phosphorylation-dependent activation. The phosphorylation sites required for lipase activity (Tyr783 in PLC-γ1 and Tyr759 in PLC-γ2) are located within this array (*Gresset et al., 2010*; *Kim et al., 1991*; *Nishibe et al., 1990*). The array also harbors two SH2 domains, nSH2 and cSH2, which mediate binding of tyrosine kinases (*Nishibe et al., 1990*) and autoinhibition of lipase activity (*Gresset et al., 2010*), respectively. Indeed, deletion or mutation of the cSH2 domain is sufficient to constitutively activate PLC-γ1 in vitro and in cells (*Gresset et al., 2010*; *Hajicek et al., 2013*). Importantly, the release of autoinhibition mediated by the cSH2 domain is coupled directly to the phosphorylation of Tyr783 (*Gresset et al., 2010*). A PLC-γ1 peptide encompassing phosphorylated Tyr783 (pTyr783) binds to the isolated cSH2 domain with high affinity (K$_D$ ~360 nM). In the context of the full-length isozyme, mutation of the cSH2 domain such that it cannot engage phosphorylated tyrosines ablates kinase-dependent activation of PLC-γ1 in vitro. Presumably, the intramolecular interaction between pTyr783 and the cSH2 domain drives a substantial conformational rearrangement resulting in activation of the enzyme. In addition to these regulatory components, the array also harbors scaffolding properties mediated by a split pleckstrin homology (sPH) domain and an SH3 domain that bind multiple signaling and adaptor proteins including the monomeric GTPase Rac2 (*Walliser et al., 2008*) and SLP-76 (*Yablonski et al., 2001*), respectively.

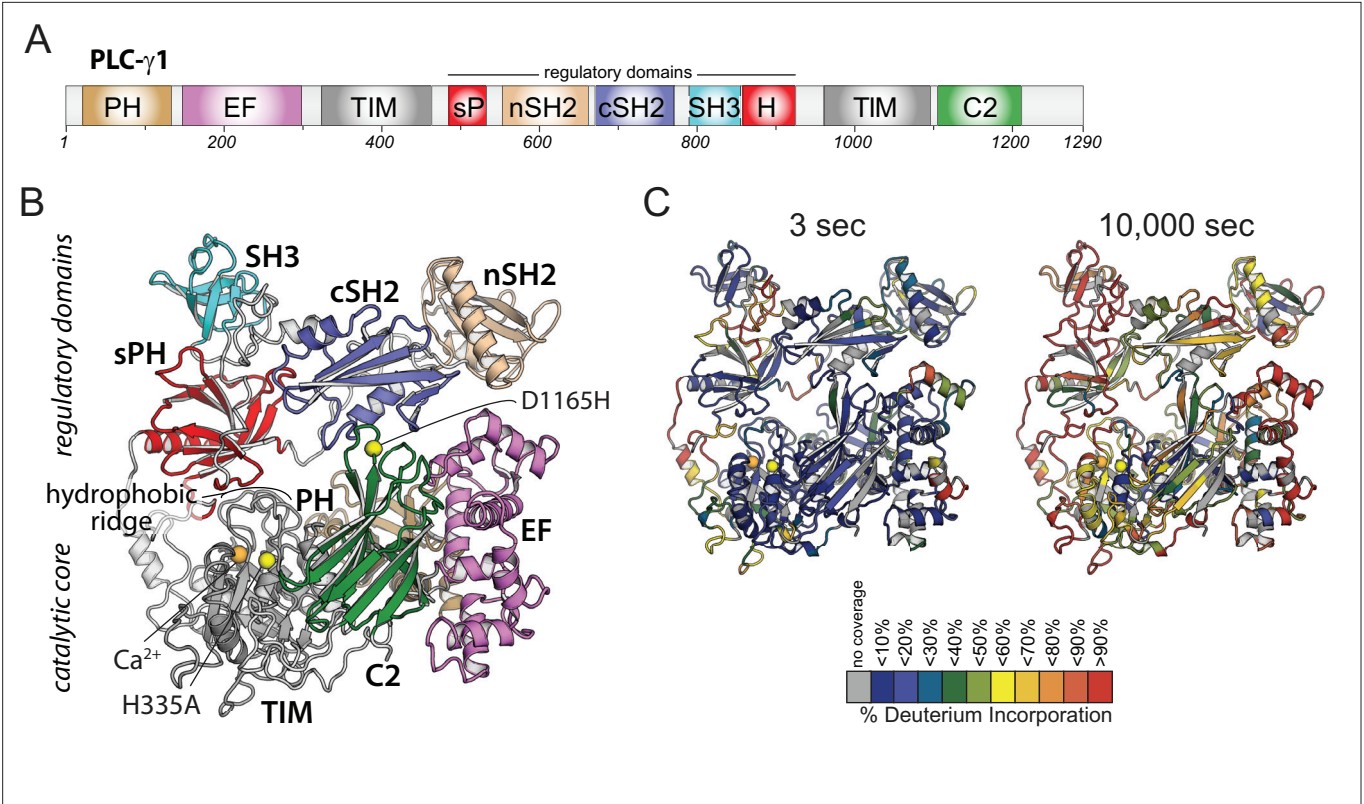

**Figure 1.** Domain architecture of phospholipase C-γ1 (PLC-γ1) and initial hydrogen-deuterium exchange mass spectrometry time course. (**A**) Domain schematic of PLC-γ1. Full-length PLC-γ1 possesses a set of regulatory domains inserted within its catalytic core. (**B**) Structure of PLC-γ1. In the basally autoinhibited state shown here (PDB ID: 6PBC), the regulatory domains prevent access to membranes by the catalytic core demarked by a Ca²⁺ cofactor (orange). His335 is also within the catalytic core and its substitution (H335A) renders the isozyme catalytically inactive (yellow). Asp1165 substitution (D1165H) disrupts the interface between the core and the regulatory array leading to a constitutively active lipase (also yellow). (**C**) Relative levels of deuterium incorporation are colored on the structure of PLC-γ1 (residue 21–1215) according to the legend at either 3 or 10,000 s of D₂O exposure.

The online version of this article includes the following source data and figure supplement(s) for figure 1:

**Source data 1.** Raw mass spectral data of deuterium incorporation per peptide.

**Source data 2.** Unprocessed images.

**Figure supplement 1.** Phospholipase C-γ1 (PLC-γ1) (H335A) is catalytically inactive while retaining capacity to bind liposomes.

---

The X-ray crystal structure of essentially full-length PLC-γ1 highlights how the regulatory array integrates these myriad functions (*Hajicek et al., 2019*) (*Figure 1A–B*). In the autoinhibited state, the array is positioned directly on top of the catalytic core of the enzyme. This arrangement, coupled with the overall electronegative character of the array effectively blocks the lipase active site from spuriously engaging the plasma membrane. Two major interfaces anchor the regulatory array to the catalytic core. The first interface is formed between the cSH2 domain and the C2 domain, and essentially buries the surface of the cSH2 domain that engages the region encompassing pTyr783. The second interface is comprised of the sPH domain and the catalytic TIM barrel. More specifically, the sPH domain sits atop the hydrophobic ridge in the TIM barrel, effectively capping the ridge and preventing its insertion into membranes, a requisite step in the interfacial activation of PLC isozymes. In contrast, the phosphotyrosine-binding pocket on the nSH2 domain, as well as the polyproline-binding surface on the SH3 domain are solvent exposed, implying that these elements are pre-positioned to efficiently engage activated RTKs and adaptor proteins, respectively. The overall architecture of the regulatory array is consistent with an enzyme that is unable to access membrane-resident substrate in the basal state. Upon activation of an RTK, e.g., FGFR1, the nSH2 domain mediates recruitment of PLC-γ1 to the phosphorylated tail of the receptor, and PLC-γ1 is subsequently phosphorylated by FGFR1 on Tyr783. pTyr783 then binds to the cSH2 domain, unlatching it from the catalytic core. This binding event is likely coupled to additional structural rearrangements of the regulatory array with respect

to the catalytic core that ultimately allows the active site to access $PIP_2$ embedded in membranes (*Hajicek et al., 2019*; *Liu et al., 2020*).

Although the core aspects of autoinhibition and phosphorylation-dependent regulation are relatively well established, less is known about the chronology of events subsequent to receptor binding that result in lipase activation. A comparison of the structure of autoinhibited PLC-γ1 and a structure of the tandem SH2 domains of PLC-γ1 bound to the phosphorylated kinase domain of FGFR1 (*Bae et al., 2009*) suggests a cogent sequence of events. In autoinhibited PLC-γ1, kinase engagement by the nSH2 domain would partially expose the surface of the cSH2 domain that binds pTyr783. Binding of pTyr783 would then fully displace the cSH2 domain from the catalytic core, presumably initiating and propagating the larger structural rearrangements that disrupt the remainder of the autoinhibitory interface, culminating in membrane engagement. Cooperativity between nSH2 and cSH2 domains of PLC-γ1 has been described previously (*Bunney et al., 2012*); however, the functional consequences of this cooperativity were linked to a reduction in the affinity for FGFR1 with no corresponding effect on lipase activity.

Here, we use hydrogen-deuterium exchange mass spectrometry (HDX-MS) to define changes in PLC-γ1 dynamics upon engagement of $PIP_2$-containing liposomes and the phosphorylated kinase domain of FGFR1. PLC-γ1 was largely quiescent in the presence of liposomes. In contrast, rapid and widespread changes in exchange kinetics were observed in a PLC-γ1-FGFR1 complex. These changes were centered on the nSH2 domain and also encompassed large swaths of the autoinhibitory interface. Changes in this exchange profile were further amplified in PLC-γ1 bound to both FGFR1 and liposome. These results are consistent with in vitro lipase assays demonstrating that engagement of FGFR1 was sufficient to elevate PLC-γ1 activity, and suggest a model in which receptors and membranes act together to prime the lipase for full activation.

## Results

### Widespread changes in the hydrogen-deuterium exchange of PLC-γ1 by kinase and liposomes

In order to assess the dynamics of PLC-γ1 using HDX-MS, it was first necessary to establish optimized conditions for protein coverage, a suitable time course of H/D exchange and constructs suitable for membrane binding. These parameters were addressed in two steps. First, optimization of HDX-MS conditions allowed for the generation of 254 peptides covering ~92% of the PLC-γ1 sequence, with an average peptide length of 13 amino acids. H/D exchange was carried out at five time points (3, 30, 300, 3000, and 10,000 s), which allowed for the interrogation of differences in both highly dynamic and stable regions of secondary structure (*Figure 1C*). Second, it was necessary to design and ensure a system that would allow PLC-γ1 to interact with $PIP_2$-containing liposomes over the time course of the exchange reaction without the catalytic conversion of the liposomes. This outcome was accomplished by introduction of a single substitution (H335A) into PLC-γ1 of a catalytic residue necessary for $PIP_2$ hydrolysis (*Figure 1B*). Although several substitutions of active site residues are known to render PLCs catalytically inactive (*Ellis et al., 1998*), His335 was specifically chosen for its role in the enzymatic process. More specifically, based on analogy to PLC-δ1, His335 is predicted to coordinate the incoming water molecule required for $PIP_2$ hydrolysis; it is predicted to be required during the transition state but should have little effect on the affinity of PLC-γ1 for $PIP_2$ as a substrate. In addition, His335 does not ligate the calcium ion cofactor in the crystal structure of PLC-γ1 and for this reason is unlikely to perturb the overall structure of the active site.

Consequently, PLC-γ1 (H335A) was purified and its lack of catalytic activity verified using a fluorescent substrate embedded within liposomes (*Figure 1—figure supplement 1*). In addition, a flotation assay was used to show that PLC-γ1 (H335A) retains capacity to associate with $PIP_2$-containing liposomes (*Figure 1—figure supplement 1*). For flotation measurements, proteins were initially treated to chelate calcium and free calcium concentrations were maintained below 400 nM to prevent substantial $PIP_2$ hydrolysis by WT PLC-γ1.

Once we determined PLCγ-1 was amenable to studies by HDX-MS, changes in the hydrogen-deuterium exchange profile of PLC-γ1 (H335A) were measured in response to either: (i) $PIP_2$-containing lipid vesicles (90% PE/10% $PIP_2$), (ii) the kinase domain of FGFR1 (FGFR1K) phosphorylated at Tyr766, or (iii) both vesicles and phosphorylated kinase (*Figure 2*). Tyrosine 766 is a major site of phosphorylation

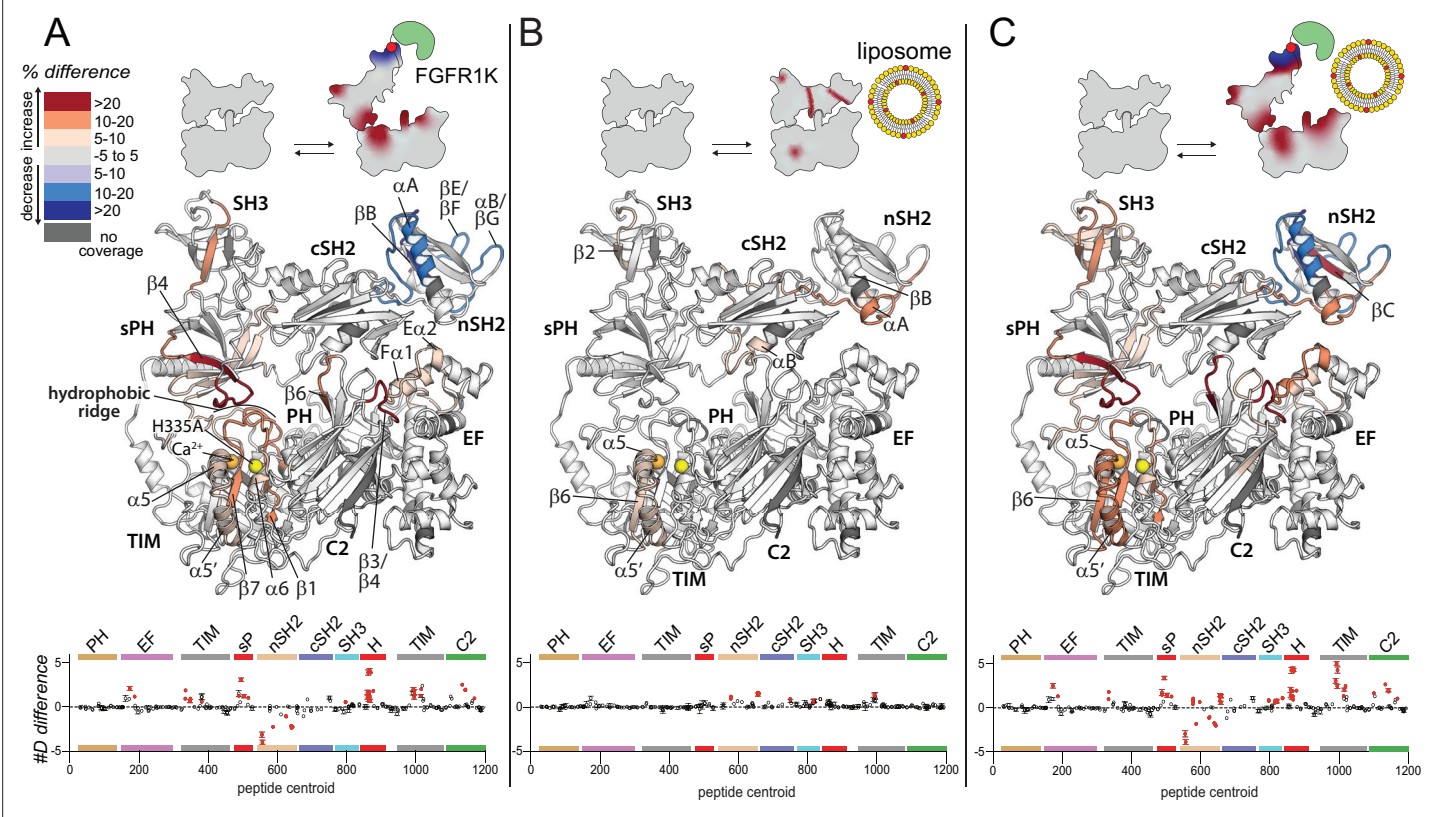

**Figure 2.** Widespread changes in deuterium exchange of phospholipase C-γ1 (PLC-γ1) upon binding kinase domain of fibroblast growth factor receptor (FGFR1K). Significant differences in deuterium incorporation are mapped on the structure of PLC-γ1 (H335A) according to the legend for the following three states: alone versus bound to phosphorylated kinase domain of fibroblast growth factor receptor (FGFR1K) (**A**), alone versus in the presence of liposomes containing phosphatidylethanolamine:phosphatidylinositol 4,5-bisphosphate (90:10) (**B**), or alone versus bound to both pFGFR1K and liposomes (**C**). In addition, *Figure 2—figure supplement 4* shows differences in exchange of PLC-γ1 (H335A) bound to either liposomes or pFGFR1K relative to a final state with PLC-γ1 (H335A) bound to both liposomes and pFGFR1K. Significant differences in any peptide required three specific conditions: greater than both a 5% and 0.4 Da difference in exchange at any time point (3, 30, 300, 3000, and 10,000 s), and a two-tailed, unpaired t-test of p<0.01. The #D difference for each condition is graphed below, which shows the total difference in deuterium incorporation over the entire hydrogen-deuterium exchange time course, with each point indicating a single peptide (error shown as SD [n=3]). Each circle represents the central residue of a corresponding peptide with the full deuterium exchange information for all peptides available in the source data. Individual peptides with a significant difference as defined above are colored red.

The online version of this article includes the following figure supplement(s) for figure 2:

**Figure supplement 1.** Kinase domain of fibroblast growth factor receptor (FGFR1K) specifically phosphorylated on Tyr766.

**Figure supplement 2.** Stable complexes of phospholipase C-γ1 (PLC-γ1) and phosphorylated kinase domain of fibroblast growth factor receptor (FGFR1K).

**Figure supplement 3.** Potential mechanism of priming of phospholipase C-γ1 (PLC-γ1) by kinase domain of fibroblast growth factor receptor (FGFR1K).

**Figure supplement 4.** Kinase domain of fibroblast growth factor receptor (FGFR1K) and liposomes act essentially independently to affect sites of exchange within phospholipase C-γ1 (PLC-γ1).

in FGFR1 and phosphorylated Tyr766 directly binds PLC-γ1 to recruit it to membranes in cells (*Eswarakumar et al., 2005*; *Mohammadi et al., 1991*). Phosphorylated Tyr766 of FGFR1K is also the major site of interaction between the kinase domain and the tandem SH2 array of PLC-γ1 in a crystal structure of these fragments in complex (*Bae et al., 2009*). Consequently, the kinase domain of FGFR1 was mutated to remove minor sites of tyrosine phosphorylation and to render the kinase resistant to phosphorylation-dependent activation before phosphorylation of Tyr766 and formation of a stable complex with PLC-γ1 (H335A) (*Figure 2—figure supplement 1*, *Figure 2—figure supplement 2*).

## FGFR1K potentially disrupts autoinhibition

There were widespread, significant differences in H/D exchange that occurred in PLC-γ1 (H335A) upon binding phosphorylated FGFR1K (*Figure 2A*). For all reported significant changes in H/D exchange, three specific conditions had to be met: greater than both a 5% and 0.4 Da difference in exchange at any time point, and a two-tailed, unpaired t-test of p<0.01. The full HDX data, and differences are reported in the source data.

Only the nSH2 domain of PLC-γ1 showed significantly decreased exchange upon binding FGFR1K. It is this domain that directly engages phosphorylated Tyr766 in the crystal structure of the kinase domain of FGFR1 bound to the tandem SH2 domains of PLC-γ1 (*Figure 2—figure supplement 3*; *Bae et al., 2009*). Consequently, the most straightforward interpretation of the reduced exchange in the complex is that it reflects the major sites of interaction between phosphorylated FGFR1 and PLC-γ1. In fact, regions of reduced exchange within the nSH2 domain match exceptionally well to those regions that directly interact with the phosphorylated portion of FGFR1K in the crystal structure. These regions of the nSH2 domain include the N-terminal portion of the αA helix and the C-terminal portion of the βB strand that interact directly with phosphorylated Tyr766 as well as the βE/βF and αB/βG loops that interact with regions of FGFR1K proximal to phosphorylated Tyr766 (*Figure 2A*, *Figure 2—figure supplement 3*, *Figure 2—figure supplement 4*). These data are consistent with previously reported differences in H/D exchange in PLC-γ1 upon binding FGFR1K (*Liu et al., 2020*).

In contrast to decrease exchange localized to the nSH2 domain, exchange was significantly increased throughout much of PLC-γ1 (H335A) upon binding FGFR1K (*Figure 2A*, *Figure 2—figure supplement 4*). In particular, increased exchange was widespread throughout the interface between the regulatory array and the catalytic core required for autoinhibition of lipase activity. For example, on one side of the interface, all of the loops within the TIM barrel that comprise the hydrophobic ridge showed increased exchange. This increased exchange was mirrored on the other side of the interface by widespread increases in exchange of most of the split PH domain. In fact, the entire first half of the sPH domain along with the first strand (β4) of the second half of the sPH domain showed increased exchange. Indeed, some of the largest increases in percent exchange were found within the β4 strand and the immediately preceding loop that interacts with the hydrophobic ridge of the TIM barrel in the autoinhibited form of PLC-γ1. Loops of the C2 domain interacting with the cSH2 domain form another important part of the autoinhibitory interface and two of these loops (β3/β4, β5/β6) within the C2 domain also showed heightened hydrogen-deuterium exchange. More peripheral regions of exchange include portions of the first two EF hands that directly support the β3/β4 loop of the C2 domain.

As described earlier, FGFR1K principally interacts with the nSH2 domain of PLC-γ1 (*Bae et al., 2009*; *Chattopadhyay et al., 1999*; *Ji et al., 1999*) such that many of the sites of increased exchange within PLC-γ1 are distant from the main binding site of FGFR1K. Consequently, while it is undoubtedly the case that the binding of FGFR1K to PLC-γ1 leads to widespread increases in the hydrogen-deuterium exchange kinetics of PLC-γ1, it is not readily apparent how FGFR1K promotes this increased exchange. Nonetheless, the increased exchange throughout the autoinhibitory interface is indicative of its increased mobility, suggesting that kinase engagement is sufficient to loosen the autoinhibitory interface and prime PLC-γ1 for more complete activation upon its phosphorylation by the kinase. This idea has been suggested previously (*Hajicek et al., 2019*).

In contrast to the widespread changes in the exchange profile of PLC-γ1 (H335A) upon engagement of phosphorylated FGFR1K, PIP$_2$-containing liposomes incubated with PLC-γ1 (H335A) produced more limited differences that were mostly restricted to the periphery of the lipase (*Figure 2B*, *Figure 2—figure supplement 4*). Overall, there were no regions of significantly decreased exchange and only three, discrete, non-contiguous regions of significantly increased exchange. Two of the regions of increased exchange were restricted to relatively small portions of either the SH3 domain (β2) or the TIM barrel (β6/α5/α5'). A larger region of exchange manifested within the two SH2 domains and encompasses regions of the nSH2 domain (αA/βB) near the site of interaction with phosphorylated Tyr766 of FGFR1K, the linker between the two SH2 domains, and the terminal portion of the cSH2 domain (αB/βG) required for autoinhibition through interactions with the C2 domain. This third, larger region is compelling since it suggests a propensity of the nSH2 domain to interact with membranes and place it within proximity of phosphorylated receptors such as FGFR1 in order to facilitate the docking of PLC-γ1 with active transmembrane receptors. It also suggests that membranes impact

regions between the two SH2 domains as well as within the cSH2 domain to possibly facilitate release of autoinhibition. Nonetheless, the overall magnitudes of differences in exchange in response to liposomes are small compared to the differences observed upon FGFR1K engagement (*Figure 2A–B*, *Figure 2—figure supplement 4*). This result suggests multiple points of transient engagement of PLC-γ1 with liposomes or cellular membranes in the absence of transmembrane binding partners. This scenario is also consistent with the tight, basal autoinhibition of PLC-γ1 in cells in the absence of a relevant, activated transmembrane receptor (*Gresset et al., 2010*). However, in the presence of relevant activated receptors as represented by phosphorylated FGFR1, the interaction of membranes with PLC-γ1 may help PLC-γ1 to engage receptors and may subsequently cooperate with receptors to facilitate release of lipase autoinhibition.

Cooperation between FGFR1 and membranes in the regulation of PLC-γ1 is also consistent with the hydrogen-deuterium exchange pattern of PLC-γ1 observed in the presence of both FGFR1K and liposomes (*Figure 2C*, *Figure 2—figure supplement 4*). Overall, similar patterns of differences in exchange are observed for the same regions of PLC-γ1 affected by either component alone. Indeed, the overlap of the exchange profiles is rather striking and is a testament to the reproducibility of the experimental technique. However, when both FGFR1K and liposomes were present, there were enhanced increases in exchange in PLC-γ1 compared to either condition alone. These increases manifest primarily within the SH3 domain and the catalytic TIM barrel. Overall, HDX-MS data suggest a modest, synergistic cooperation between FGFR1K and liposomes to increase the conformational flexibility of the regulatory domains relative to the catalytic core of PLC-γ1.

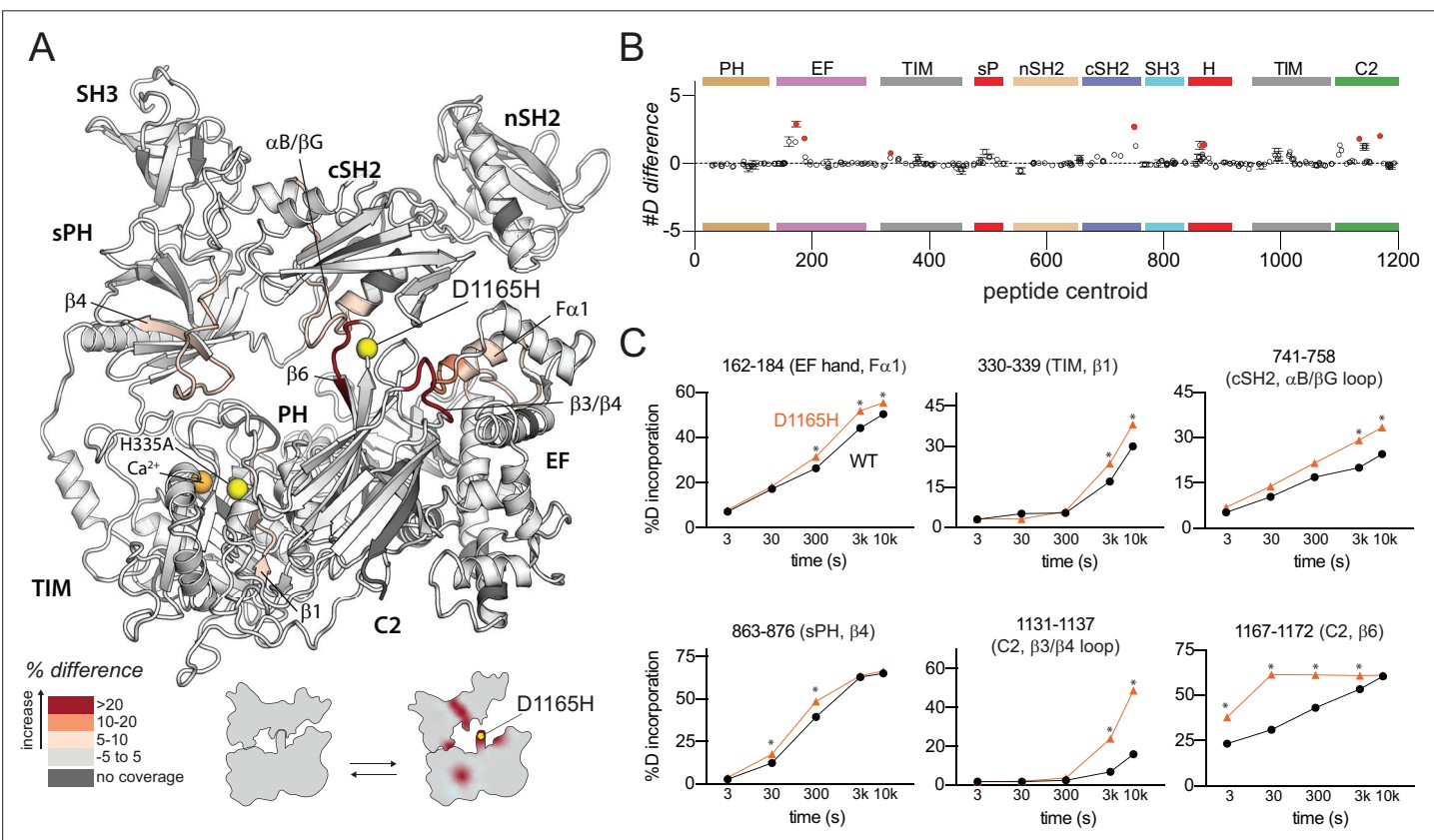

**Figure 3.** Oncogenic substitution of phospholipase C-γ1 (PLC-γ1) mimics kinase engagement. (**A**) Significant differences in deuterium incorporation for PLC-γ1 (H335A) versus the oncogenic mutant of PLC-γ1 (H335A+D1165H) were determined and mapped on the structure of PLC-γ1. (**B**) The #D difference upon mutation shows the total difference in deuterium incorporation over the entire hydrogen-deuterium exchange time course, with each point indicating a single peptide (error shown as SD [n=3]). Individual peptides with a significant difference between conditions (defined as greater than both a 5% and 0.4 Da difference in exchange at any time point, and a two-tailed, unpaired t-test of p<0.01) are colored red. (**C**) A selection of peptides showing significant differential exchange at any time point (data shown as mean ± SD [n=3]), asterisks indicate significant time points for peptides as defined above. Full deuterium exchange data is available in the source data.

## Oncogenic substitution of PLC-γ1 mimics engagement by kinase and liposomes

The PLC-γ1 interface between its regulatory domains and catalytic core is frequently mutated in certain cancers and immune disorders (*Koss et al., 2014*). We have previously proposed that these mutations likely disrupt this interface leading to elevated, basal phospholipase activity (*Hajicek et al., 2019*). If true, oncogenic substitutions at this interface in PLC-γ1 might recapitulate changes in deuterium exchange in PLC-γ1 observed upon binding phosphorylated FGFR1K. This idea was tested using HDX-MS and PLC-γ1 (H335A) harboring an additional substitution (D1165H) frequently found in leukemias and lymphomas (*Kataoka et al., 2015*) (*Figure 3*). To make sure that there were no complications from the use of the H335A mutant, we also carried out HDX-MS experiments comparing D1165 versus D1165H in a catalytically active background (H335), which showed very similar differences in exchange upon mutation of D1165 (*Figure 1—source data 1*).

Asp1165 resides in a loop between strands β5 and β6 of the C2 domain where it is required to stabilize extensive interactions between the C2 and cSH2 domains. These interactions are required to maintain PLC-γ1 in an autoinhibited state since the basal activity of PLC-γ1 (D1165H) is ~1500-fold higher than the WT phospholipase (*Hajicek et al., 2019*).

In comparison to catalytically inactive PLC-γ1 harboring H335A, the same version of PLC-γ1 with the addition of D1165H exhibited increased deuterium exchange throughout the interface between the regulatory domains and the catalytic core (*Figure 3*). In the immediate vicinity of D1165H, deuterium exchange increased within the β6 strand of the C2 domain suggesting that Asp1165 serves to support the position of the β6 strand. This idea is consistent with previous molecular dynamics simulations of PLC-γ1 that highlighted an unraveling of the β6 strand upon introduction of D1165H (*Hajicek et al., 2019*). The previous molecular dynamics simulations also highlighted a relatively large (~10 Å) movement of the cSH2 upon collapse of the β6 strand, and this movement may be reflected in the increased exchange of the nearby αB helix and αB/βG loop of the cSH2 domain. Other regions of increased exchange are more distant and include portions of the first EF hand, sPH domain, and TIM barrel. Overall, regions of increased exchange in catalytically inactive PLC-γ1 (D1165H) overlap well with regions of increased exchange in PLC-γ1 (H335A) when it engages phosphorylated FGFR1K (*Figure 2A*). This comparison further supports the notion that disengagement of the catalytic core from the regulatory domains will enhance lipase activity, with oncogenic mutations and binding to phosphorylated FGFR1K mediating this process. Indeed, it should be noted that regions of increased exchange are more extensive in the complex of PLC-γ1 and FGFR1K suggesting that oncogenic mutations cannot fully recapitulate activation by phosphorylated FGFR1K.

## Oncogenic substitution uncovers functional cooperativity within PLC-γ1

Constitutively active variants of PLC-γ1 and PLC-γ2 have untapped reserves of lipase activity. For example, although PLC-γ1 (D1165H) possesses exceptionally high basal lipase activity, this activity can be further enhanced in cells by the epidermal growth factor receptor suggesting that normal cellular regulation is partially preserved in constitutively active forms of PLC-γ1 (*Hajicek et al., 2019*). To test this idea, changes in the hydrogen-deuterium exchange profile of PLC-γ1 (H335A+D1165H) were determined in response to phosphorylated FGFR1K, liposomes, or both (*Figure 4*, *Figure 4—figure supplement 1*, *Figure 4—figure supplement 2*). In comparison to the identical measurements described earlier using PLC-γ1 (H335A), these measurements are particularly revealing for several reasons.

First, phosphorylated FGFR1K produced almost identical changes in the exchange profiles of the two forms of PLC-γ1 as highlighted in the comparison of *Figures 2A and 4A* (see also *Figure 4—figure supplement 1*). Not only does this result further attest to the reproducibility of the experimental design, it also strongly suggests that WT and constitutively active forms of PLC-γ1 respond similarly to engagement by kinases.

Second, and in contrast to effects mediated by phosphorylated FGFR1K, PIP$_2$-containing liposomes produced much more extensive and widespread changes in the hydrogen-deuterium exchange profile of mutant PLC-γ1 (H335A+D1165H) (*Figure 4B*, *Figure 4—figure supplement 1*) compared to PLC-γ1 (H335A) (*Figure 2B*). For example, while no portion of the EF hands of PLC-γ1 (H335A) showed differences in exchange upon addition of liposomes, the entire first EF hand (Eα1/Fα1) as well as the start of the second EF hand (Eα2) of PLC-γ1 (H335A+D1165H) showed increased exchange. In

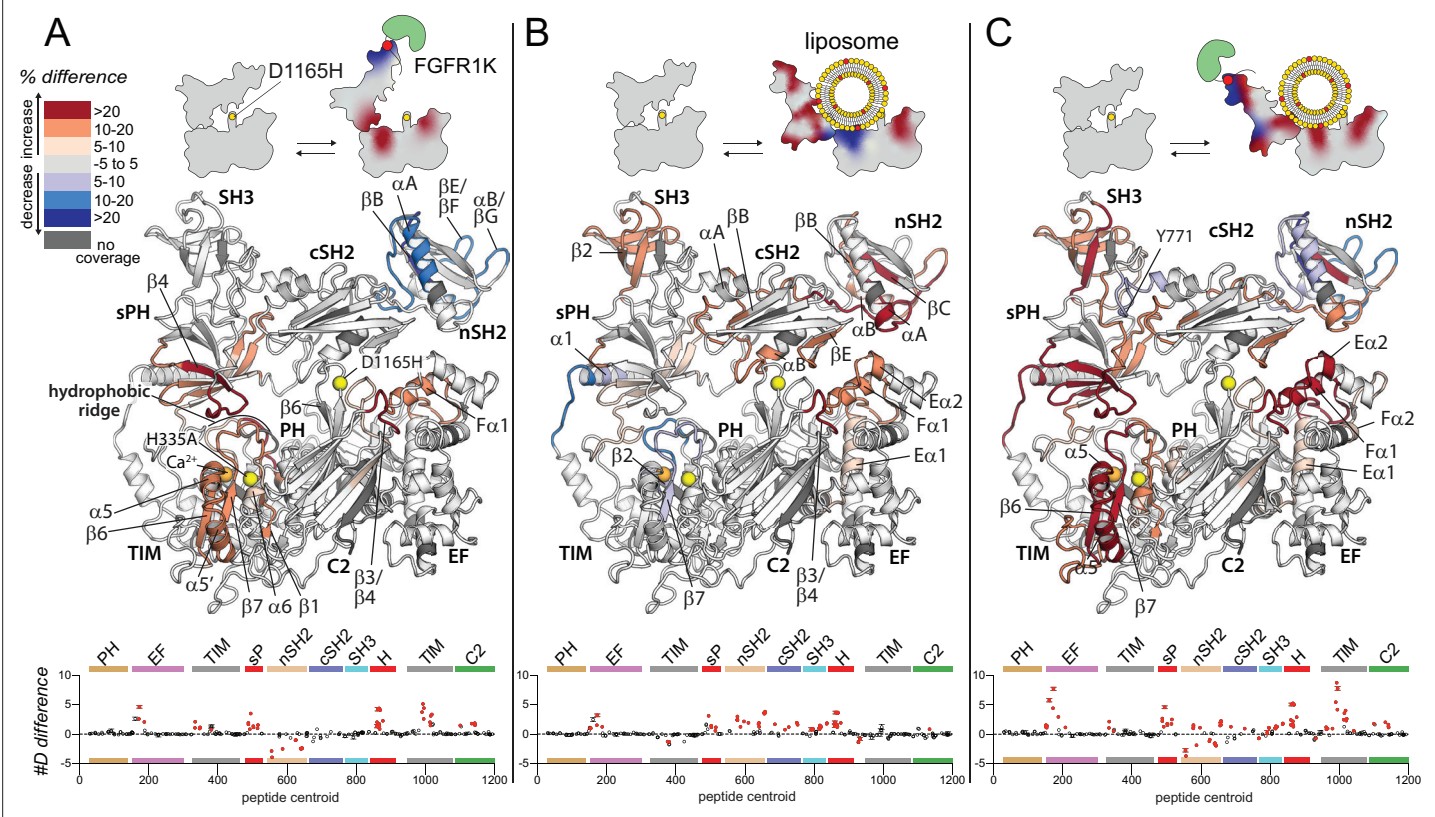

**Figure 4.** Oncogenic substitution uncovers functional cooperativity within phospholipase C-γ1 (PLC-γ1). Differences in deuterium incorporation were measured for PLC-γ1 (H335+D1165H) in three states: alone versus bound to phosphorylated kinase domain of fibroblast growth factor receptor (pFGFR1K) (**A**), alone versus in the presence of phosphatidylethanolamine:phosphatidylinositol 4,5-bisphosphate (90:10) liposomes (**B**), or alone versus with both pFGFR1K and liposomes present (**C**). In addition, *Figure 4—figure supplement 1* shows differences in exchange of PLC-γ1 (H335A+D1165H) bound to either liposomes or pFGFR1K relative to a final state with PLC-γ1 (H335A+D1165H) bound to both liposomes and pFGFR1K. Differences in deuterium exchange for a series of time points (3, 30, 300, 3000, and 10,000 s) were calculated relative to PLC-γ1 (H335A+D1165H) alone and peptides with significant differential exchange at any time point (both a 5% and 0.4 Da difference in exchange at any time point, and a two-tailed, unpaired t-test of p<0.01) mapped onto the structure. The #D difference for each condition is graphed below, which shows the total difference in deuterium incorporation over the entire hydrogen-deuterium exchange time course, with each point indicating a single peptide (error shown as SD [n=3]); peptides with significant differences between conditions as defined above are red.

The online version of this article includes the following figure supplement(s) for figure 4:

**Figure supplement 1.** Kinase domain of fibroblast growth factor receptor (FGFR1K) and liposomes cooperate to affect the deuterium exchange of phospholipase C-γ1 (PLC-γ1) (D1165H).

**Figure supplement 2.** Kinase domain of fibroblast growth factor receptor (FGFR1K) and liposomes have enhanced capacity to affect the deuterium exchange of the oncogenic mutant of phospholipase C-γ1 (PLC-γ1) compared to the wild type (WT).

addition, the adjacent β3/β4 loop of the C2 domain had robustly increased exchange upon membrane binding in the D1165H variant, whereas this region in PLC-γ1 (H335A) was essentially unresponsive to lipids. In the same vein, while liposomes elicited only relatively modest and discrete increases in the exchange of the regulatory domains of PLC-γ1 (H335A), under the same conditions, there were widespread and robust increases in exchange within the regulatory domains of PLC-γ1 (H335A+D1165H). These regions include the entire first half of the sPH domain, the majority of both the nSH2 and SH3 domains, as well as the EF and BG loops (βE/βF, αB/βG) of the cSH2 domain critical for autoinhibition of lipase activity.

Third, and perhaps most intriguing, while liposomes failed to decrease exchange of any part of PLC-γ1 (H335A), under identical conditions, several portions of PLC-γ1 (H335A+D1165H) exhibited decreased exchange. These regions of decreased exchange are primarily in the TIM barrel: β2-β2' and β7 strands as well as the subsequent loops comprising the majority of the hydrophobic ridge. The hydrophobic ridge is generally assumed to insert into membranes during interfacial catalysis and the

decreased exchange of this region may indicate its direct interaction with liposomes during aborted attempts at catalysis. A second region of decreased exchange encompasses α1 of the sPH domain as well as the adjoining linker connecting to the second half of the TIM barrel. This linker is disorderd in the crystal structure of full-length PLC-γ1 and it is intriguing to speculate that this region may act as a hinge that allows the regulatory array to move in order to allow access of liposomes to the hydrophobic ridge.

Finally, the exchange profile of PLC-γ1 (H335A+D1165H) upon the addition of both phosphorylated FGFR1K and liposomes (*Figure 4C*) approximately resembles the composite profile of each component alone, with two interesting caveats. First, a portion of the linker between the cSH2 and SH3 domains that includes frequently phosphorylated Tyr771 possesses decreased exchange that cannot be accounted by a composite profile. This difference presumably reflects structural changes in PLC-γ1 (H335A+D1165H) that arise due to cooperative interactions among the PLC isozyme, FGFR1K, and liposomes. This presumption is reinforced by the second caveat: in the presences of both FGFR1K and liposomes, there is a striking synergy in the differences in exchange for PLC-γ1 (H335A+D1165H) compared to PLC-γ1 (H335A), with liposomes enhancing the changes observed with only FGFR1K approximately two-fold (*Figure 4C*, **lower panel**). These regions include most of the first and second

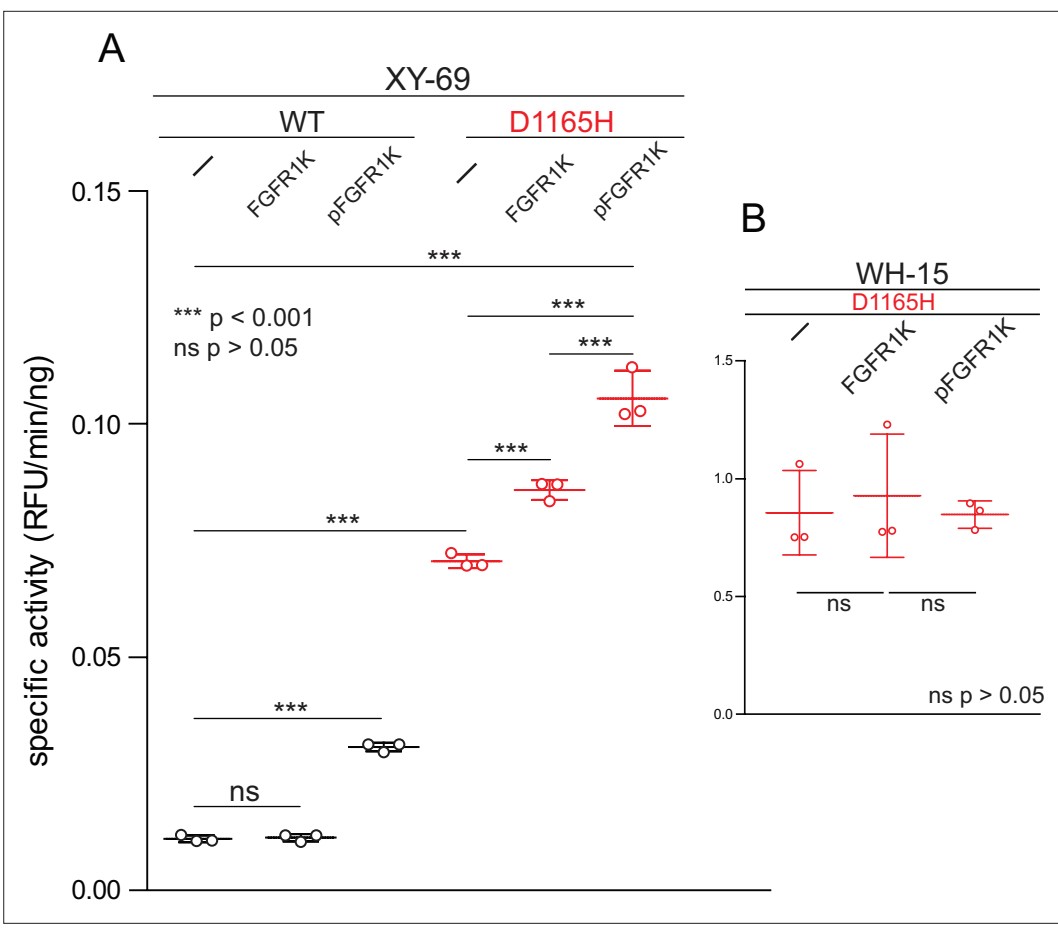

**Figure 5.** Phosphorylated kinase domain of fibroblast growth factor receptor (FGFR1K) increases phospholipase C-γ1 (PLC-γ1) specific activity. (**A**) Specific activities measured with the membrane-embedded substrate XY-69. XY-69 (0.5 μM) was incorporated into liposomes comprised of phosphatidylethanolamine:phosphatidylinositol 4,5-bisphosphate (80:20) prior to addition of indicated variants of PLC-γ1 (10 nM WT; 0.3 nM D1165H). Specific activities were calculated for PLC-γ1 alone and in the presence of either unphosphorylated FGFR1K or its phosphorylated counterpart (pFGFR1K). (**B**) Specific activities of PLC-γ1 (D1165H) measured with water-soluble WH-15 (5 μM). In all cases, specific activities are presented as the mean ± SD of three independent experiments (n=3), each with three or more technical replicates. Statistical significance was determined with a two-tailed, unpaired t-test.

EF hands as well as a large segment of the second half of the TIM barrel spanning strands β6 and β7 and including helices α5 and α5'.

Overall, the exchange results indicate that the oncogenic substitution, D1165H, leads to a much more conformationally flexible PLC-γ1 that more readily interacts with lipid bilayers, possibly in a fashion that mirrors interfacial catalysis by PLC isozymes. This process would require a large movement by the regulatory domains as suggested by the icons in *Figure 4*.

## Phosphorylated FGFR1K increases PLC-γ1 specific activity

The HDX kinetics described above strongly suggest that the initial engagement of PLC-γ isozymes by kinases may be sufficient to enhance phospholipase activity irrespective of subsequent phosphorylation. This idea was tested using XY-69 (*Huang et al., 2018*), a fluorescent analog of PIP₂, embedded in lipid vesicles (*Figure 5*). PLCs readily hydrolyze XY-69 leading to increased fluorescence that can be followed in real time. As shown previously, when XY-69 in lipid vesicles is mixed with PLC-γ1, the specific activity of the phospholipase is low, indicative of basal autoinhibition. This activity remained unchanged upon pre-incubation of PLC-γ1 with catalytically inactive FGFR1K that was not phosphorylated. In contrast, when PLC-γ1 was pre-incubated with catalytically inactive FGFR1K phosphorylated at Tyr766 to promote complex formation with PLC-γ1, the specific lipase activity of PLC-γ1 increased approximately three-fold. These results indicate that kinase engagement by PLC-γ1 is sufficient to enhance lipase activity and supports the proposition that PLC-γ1 is allosterically modulated by FGFR1K upon complex formation and irrespective of the phosphorylation state of the lipase.

Similar results were observed with PLC-γ1 (D1165H). In particular, addition of catalytically inactive, phosphorylated FGFR1K increased the basal lipase activity of PLC-γ1 (D1165H) approximately two-fold. The basal capacity of PLC-γ1 (D1165H) to hydrolyze vesicle-embedded XY-69 was substantially higher than the equivalent activity of WT PLC-γ1, but this result has been previously published (*Huang et al., 2018*) and comports with the higher constitutive activity of PLC-γ1 (D1165H) in cells (*Patel et al.,*

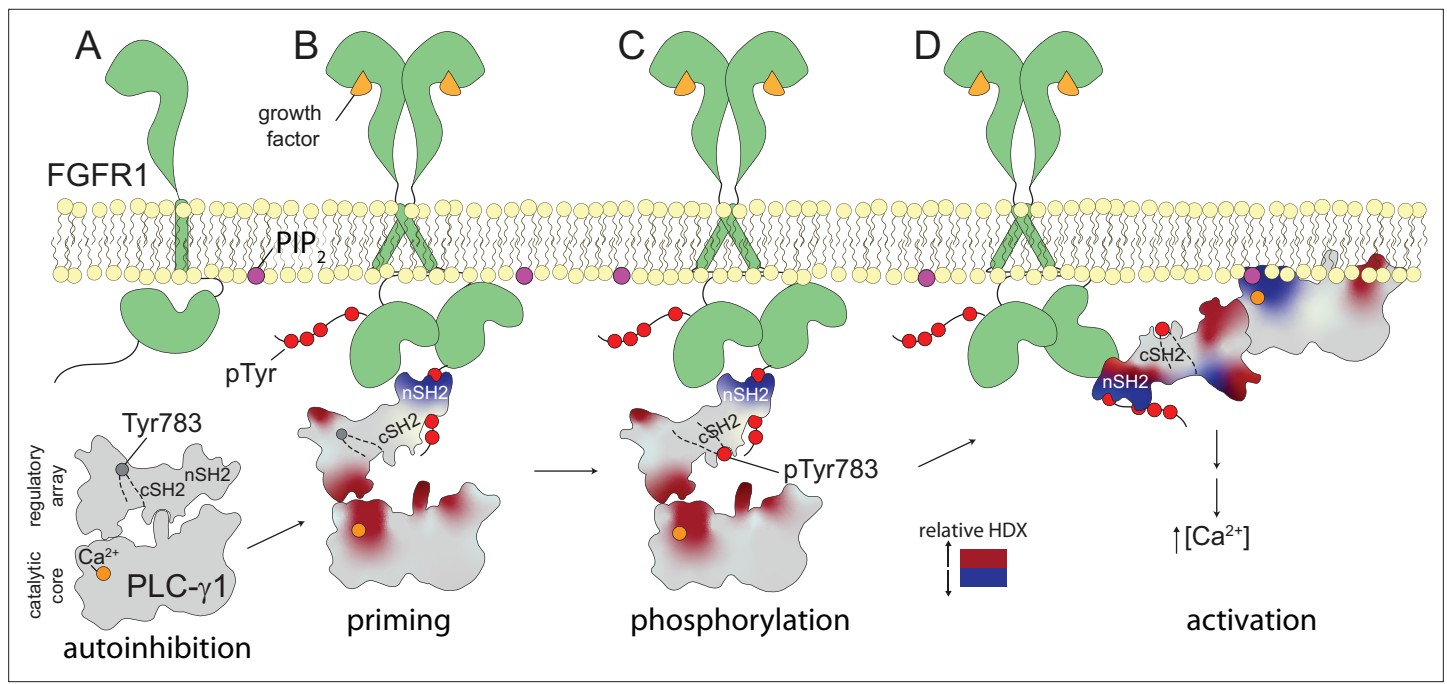

**Figure 6.** Multi-step activation of phospholipase C-γ1 (PLC-γ1). Initially, fibroblast growth factor receptor (FGFR1) is monomeric and inactive while PLC-γ1 is cytosolic and basally autoinhibited (**A**). Growth factor-promoted dimerization of FGFR1 leads to its tyrosine phosphorylation and the recruitment of PLC-γ1 through interactions mediated by the nSH2 domain of PLC-γ1 (**B**). Complex formation reduces deuterium exchange within the nSH2 domain while simultaneously increasing deuterium exchange throughout the interface between the catalytic core and regulatory domains of PLC-γ1. Increased deuterium exchange is consistent with a loosening of the autoinhibited form of PLC-γ1 shown schematically. Active FGFR1 subsequently phosphorylates Tyr783 of PLC-γ1 to favor the interaction of pTyr783 and the cSH2 domain of PLC-γ1 (**C**). This step reinforces the disruption of the interface between regulatory and catalytic domains to favor a fully open form of PLC-γ1 capable of productively engaging membranes and manifesting as a decrease in the deuterium exchange of the catalytic TIM barrel (**D**).

*2020*). Interestingly, the addition of catalytically inactive but non-phosphorylated FGFR1K also led to a significant increase in the specific activity of PLC-γ1 (D1165H). We are unable to readily account for this observation, but it should be noted that the crystal structure of phosphorylated FGFR1K bound to the two SH2 domains of PLC-γ1 highlights a secondary interface outside of the canonical pTyr-binding site that contributes to the affinity of the complex (*Bae et al., 2009*). Therefore, unphosphorylated FGFR1K may have sufficient affinity for PLC-γ1 that it is able to enhance the lipase activity of constitutively active PLC-γ1 (D1165H) but not the more basally autoinhibited, WT form.

Autoinhibition of the PLC-γ isozymes presumably arises due to steric hindrance by the regulatory array preventing the catalytic core from productively engaging lipid membranes to hydrolyze XY-69. In contrast, WH-15 is a soluble, fluorescent substrate of PLCs that is insensitive to the autoregulation of the PLC-γ isozymes (*Huang et al., 2011*). As expected, the hydrolysis of WH-15 by PLC-γ1 (D1165H) is insensitive to either form of FGFR1K (*Figure 5B*), and these results further support the idea that the complex of FGFR1K and PLC-γ1 diminishes autoregulation by loosening the conformational ensemble of the autoinhibited form of PLC-γ1.

## Discussion

A general model for the regulated activation of the PLC-γ isozymes posits a conserved, catalytic core that is basally prevented from accessing its lipid substrate, $PIP_2$, when $PIP_2$ is embedded in membranes (*Figure 6*). In essence, the two-dimensional nature of the membrane bilayer is key to this regulation since soluble substrates such as $PIP_2$ in detergent bypass this regulation and freely access the solvent-accessible active site. In contrast, the active site of PLC-γ isozymes is prevented from accessing $PIP_2$ in membranes by a set of regulatory domains that must undergo conformational rearrangements before the catalytic core can dock with membranes and engage $PIP_2$. This model derives from substantial cellular (*DeBell et al., 2007*; *DeBell et al., 1999*; *Horstman et al., 1999*; *Poulin et al., 2000*; *Schade et al., 2016*), biochemical (*Bunney et al., 2012*; *Gresset et al., 2010*; *Poulin et al., 2005*), and biophysical (*Bunney et al., 2012*; *Hajicek et al., 2013*) data and is perhaps most self-evident for the high-resolution structure of autoinhibited, full-length PLC-γ1 that highlights the arrangement of the regulatory domains relative to the catalytic core that prevents productive engagement of membranes (*Hajicek et al., 2019*).

What is much less well understood is the sequence of events culminating in the conformational rearrangements required for the activation of the PLC-γ isozymes. It is universally accepted that the phosphorylation of Tyr783 in PLC-γ1 or the equivalent residue, Tyr759, in PLC-γ2 is absolutely required for regulated activation, but the steps preceding or subsequent to these phosphorylations continue to be debated. The studies presented here address these less well-understood steps and support the view of the PLC-γ isozymes as relatively quiescent lipases prior to recruitment to membranes. Indeed, a major conclusion of these studies is that recruitment to membranes appears to be integral to activation.

Two aspects of this recruitment and activation deserve special consideration. First is the engagement of PLC-γ isozymes with RTKs or co-receptors. Complex formation requires receptors or co-receptors to be pre-phosphorylated at specific tyrosines to provide docking sites for the PLC-γ isozymes. Based on the majority of previous studies that address this point, the nSH2 domain of the PLC-γ isozymes directly engages phosphorylated tyrosines in the myriad of relevant, membrane-resident receptors and co-receptors. This point is also consistent with the relatively large decreases in amide proton exchange of the nSH2 domain of PLC-γ1 upon addition of the phosphorylated kinase domain of FGFR1. However, much more interesting is the widespread increases in amide exchange of PLC-γ1 upon complex formation. These widespread increases highlight a phospholipase that is exquisitely—and indeed, globally—tuned to engagement of the kinase. The most significant implication being that engagement of the nSH2 domain by a phosphorylated receptor or co-receptor increases the dynamic flexibility of large portions of the phospholipase. This increased flexibility will tend to loosen the autoinhibited state of the phospholipase to promote activation. In essence, initial recruitment of the PLC-γ isozymes to phosphorylated receptors or co-receptors is a crucial first step for lipase activation. We previously inferred this 'priming' step based solely on structural considerations, and the exchange data presented here strongly support this idea. Indeed, since priming is predicted to disfavor the autoinhibited state, it is also predicted to favor activation. Thus a priming step—that is independent of subsequent phosphorylation of PLC-γ1—readily explains the increased lipase activity of PLC-γ1

operating on membrane-embedded $PIP_2$ and bound to the phosphorylated, but catalytically inactive, kinase domain of FGFR1.

The second aspect to consider is the intrinsic role of membranes during the activation of PLC-γ isozymes. Lipid vesicles containing $PIP_2$ were relatively ineffectual in altering the amide exchange of WT PLC-γ1. This situation comports with the tight basal autoinhibition of PLC-γ isozymes in cells where the isozymes are inevitably in close proximity to lipid membranes. In contrast, the exchange kinetics of WT PLC-γ1 bound to the phosphorylated kinase domain of FGFR1 were amplified by the addition of lipid vesicles. Amplified exchange was even more extreme for PLC-γ1 (D1165H) containing a substitution introduced to perturb the interface between the catalytic core and regulatory domains (*Hajicek et al., 2019*). These results indicate that once primed by kinase—or indeed, mutation— PLC-γ1 becomes more sensitive to the presence of lipid vesicles. The net outcome likely shifts PLC-γ1 toward conformations that can productively engage membranes to hydrolyze membrane-bound $PIP_2$.

These two aspects should not be considered in isolation. That is, the activation of PLC-γ isozymes involves the intertwined events of: (i) docking to membrane-resident receptors or co-receptors and (ii) the forced proximity at membranes of the dock complexes. The two events cooperate to initiate activation of PLC-γ isozymes (*Figure 6*), presumably by increasing accessibility to membrane-bound $PIP_2$. At this point, it should be emphasized that the experimental setup studied here does not fully recapitulate the cellular activation process and almost certainly underestimates potential cooperativity. More specifically, to facilitate experimental tractability, and especially the desire to study complex formation independent of recruitment to membranes, these studies exclusively used an isolated kinase domain of FGFR1 *not* embedded in membranes. We predict cooperativity will be greatly enhanced for receptors or co-receptors embedded in membranes as typically found in cells. Increased cooperativity will result in increased sensitivity of activation of the PLC-γ isozymes. It should also be noted that these two steps along the activation pathway are general in that they are likely to manifest for any receptor or co-receptor that recruits PLC-γ1 or PLC-γ2 to membranes through motifs that contain phosphorylated tyrosine and bind the nSH2 domain of PLC-γ isozymes.

Up to this point, we have considered only membrane recruitment and activation of PLC-γ isozymes by phosphorylated receptors and co-receptors. However, these are not the only proteins that bind PLC-γ isozymes at membranes. For example, several scaffolding proteins such as SLP-76 that contain polyproline regions bind the SH3 domain of PLC-γ isozymes (*Kim et al., 2000*; *Manna et al., 2018*; *Rouquette-Jazdanian et al., 2012*; *Tvorogov and Carpenter, 2002*). Similarly, the small GTPase, Rac2, specifically engages the sPH domain of PLC-γ2 to promote lipase activation (*Bunney et al., 2009*; *Piechulek et al., 2005*; *Walliser et al., 2008*). We do not know if these complexes affect the conformational dynamics of the PLC-γ isozymes. However, if they do, these complexes may also prime the PLC-γ isozymes for activation at membranes with one potential outcome being an intricate interplay of multiple binding events regulating the activation of the PLC-γ isozymes.

# Materials and methods

## Key resources table

| Reagent type (species) or resource | Designation | Source or reference | Identifiers | Additional information |
|---|---|---|---|---|
| Strain, strain background (*Escherichia coli*) | NEB 5-alpha | New England BioLabs | Cat# C2987I | Chemically competent cells |
| Strain, strain background (*E. coli*) | DH10Bac | Thermo Fisher Scientific | Cat# 10361012 | Chemically competent cells |
| Cell line (*Trichoplusia ni*) | HighFive | Invitrogen | Cat# B85502 RRID:CVCL_C190 | |
| Recombinant DNA reagent | pFB-LIC2-PLC-γ1 (21–1215) (plasmid) | PMID:31889510 | | Vector is a modified version of pFastBacHT1 containing His₆ tag and TEV cleavage sequence |
| Recombinant DNA reagent | pFB-LIC2-PLC-γ1(21–1215) H335A (plasmid) | This paper | | Vector is a modified version of pFastBacHT1 containing His₆ tag and TEV cleavage sequence |

*Continued on next page*

*Continued*

| Reagent type (species) or resource | Designation | Source or reference | Identifiers | Additional information |
|---|---|---|---|---|
| Recombinant DNA reagent | pFB-LIC2-PLC-γ1(21–1215) H335A, D1165H (plasmid) | This paper | | Vector is a modified version of pFastBacHT1 containing His$_6$ tag and TEV cleavage sequence |
| Recombinant DNA reagent | pUC57-Amp-FGFR1K(458-774) Y463F, Y583F, Y585F, Y730F (plasmid) | This paper | | Cloning vector synthesized by Genewiz |
| Recombinant DNA reagent | pFB-LIC2-FGFR1K(458-774) Y463F, Y583F, Y585F, Y730F (plasmid) | This paper | | Vector is a modified version of pFastBacHT1 containing His$_6$ tag and TEV cleavage sequence |
| Recombinant DNA reagent | pFB-LIC2-FGFR1K(458-774) Y463F, Y583F, Y585F, Y730F, Y653F, Y654F (plasmid) | This paper | | Vector is a modified version of pFastBacHT1 containing His$_6$ tag and TEV cleavage sequence |
| Recombinant DNA reagent | pFB-LIC2-FGFR1K(458-774) Y463F, Y583F, Y585F, Y730F, Y766F (plasmid) | This paper | | Vector is a modified version of pFastBacHT1 containing His$_6$ tag and TEV cleavage sequence |
| Chemical compound, drug | L-α-phosphatidylethanolamine (Liver, Bovine) | Avanti Polar Lipids | Cat# 840,026 | |
| Chemical compound, drug | L-α-phosphatidylinositol-4,5-bisphosphate (Brain, Porcine) (ammonium salt) | Avanti Polar Lipids | Cat# 840,046 | |
| Software, algorithm | PEAKS | Bioinformatics Solutions Inc (BSI) | | Version 7 (PEAKS7) |
| Software, algorithm | HDExaminer | Sierra Analytics | | |
| Software, algorithm | PyMOL | PyMOL by Schrödinger | RRID:SCR_000305 | Version 2.5.2 |
| Other | WH-15 | PMID:21158426 | | Fluorescent PIP$_2$ analogue, soluble |
| Other | XY-69 | PMID:29263090 | | Fluorescent PIP$_2$ analogue, membrane-associated |

## Molecular cloning

### Phospholipase C-γ1

Transfer vector (pFB-LIC2) encoding WT rat PLC-γ1 (residues 21–1215) and PLC-γ1 (D1165H) have been previously described (*Hajicek et al., 2019*). The H335A mutation that renders PLC-γ1 catalytically inactive was introduced into the transfer vector encoding WT PLC-γ1 (residues 21–1215) or PLC-γ1 (D1165H) using standard primer-mediated mutagenesis (Agilent Technologies; QuikChange site-directed mutagenesis manual). Mutagenesis was confirmed by automated DNA sequencing of the open reading frame. The PLCs encoded by these constructs are referred as PLC-γ1 (H335A) and PLC-γ1 (H335A+D1165H), respectively.

### Kinase domain of the fibroblast growth factor receptor

Plasmid DNA encoding the kinase domain (residues 458–774) of human fibroblast growth factor receptor 1 (FGFR1) with four substitutions (Y463F, Y583F, Y585F, Y730F) that eliminate sites of phosphorylation was synthesized by Genewiz (South Plainfield, NJ). The plasmid DNA encoding this variant of FGFR1K was then sub-cloned from a pUC57-Amp vector to a modified pFastBac-HT vector (pFB-LIC2) using a ligation-independent cloning strategy. The baculovirus expression vector, pFB-LIC2, incorporates a His6 tag and a tobacco etch virus (TEV) protease site at the amino terminus of the expressed protein. This transfer vector was then mutated to encode two forms of FGFR1K: (i) a version mutated (Y653F and Y654F) to prevent phosphorylation-dependent activation of kinase activity, i.e., kinase inactive, and (ii) a version that retained kinase activity but was mutated (Y766F) to prevent phosphorylation-dependent interaction with PLC-γ1. Mutations were introduced using standard primer-mediated mutagenesis (Agilent Technologies; QuikChange site-directed mutagenesis manual) and all FGFR1K variants were verified by automated DNA sequencing of the corresponding open reading frames.

## Protein expression and purification

### Phospholipase C-γ1

Expression and purification of PLC-γ1 variants were previously described (*Hajicek et al., 2019*) and used here with minor modifications. Briefly, 4 L of High Five (*Trichoplusia ni*) cells at a density

of approximately 2.0 × 10⁶ cells/mL were infected with amplified baculovirus stock (10–15 mL/L) encoding individual PLC-γ1 variants. Cells were harvested approximately 48 hr post-infection by centrifugation at 6,000 rpm in a Beckman JA-10 rotor at 4°C. Cell pellet was resuspended in 200 mL of buffer N1 (20 mM HEPES [pH 7.5], 300 mM NaCl, 10 mM imidazole, 10% v/v glycerol, and 0.1 mM EDTA) supplemented with 10 mM 2-mercaptoethanol and four EDTA-free cOmplete protease inhibitor tablets (Roche Applied Science) prior to lysis using the Nano DeBEE High Pressure Homogenizer (BEE International). Lysate was centrifuged at 50,000 rpm for 1 hr in a Beckman Ti70 rotor. The supernatant was filtered through a 0.45 μm polyethersulfone low protein-binding filter and loaded onto a 5 mL HisTrap HP immobilized metal affinity chromatography (IMAC) column equilibrated in buffer N1. The column was washed with 15 column volumes (CV) of buffer N1, followed by 15 CV of 2.5% buffer N2 (buffer N1 + 1 M imidazole). Bound proteins were eluted with 40% buffer N2. Fractions containing PLC-γ1 were pooled and dialyzed overnight in the presence of 2% (w/w) TEV protease to remove the His6 tag in a buffer solution containing 20 mM HEPES (pH 7.5), 300 mM NaCl, 10% v/v glycerol, 1 mM dithiothreitol (DTT), 1 mM EDTA. The sample was subsequently diluted two-fold with buffer N1 and applied to a 5 mL HisTrap HP column. Flow-through fractions containing cleaved PLC-γ1 were pooled, diluted four-fold with buffer Q1 (20 mM HEPES (pH 7.5) and 2 mM DTT), and loaded onto an 8 mL SourceQ anion exchange column equilibrated in 10% buffer Q2 (buffer Q1 +1 M NaCl). Bound proteins were eluted in a linear gradient of 10–60% buffer Q2 over 50 CV. Fractions containing PLC-γ1 were pooled, concentrated using a GE Healthcare VivaSpin 6 50K molecular weight cut-off (MWCO) centrifugal concentrator and applied to a 16 mm × 700 mm HiLoad Superdex 200 size exclusion column equilibrated in a buffer solution containing 20 mM HEPES (pH 7.5), 150 mM NaCl, 5% v/v glycerol, and 2 mM DTT. Pure PLC-γ1 was concentrated to a final concentration of 40–80 mg/mL, aliquoted, snap-frozen in liquid nitrogen, and stored at –80°C until use.

## Kinase domain of the fibroblast growth factor receptor
Expression and purification of kinase-inactive FGFR1K follows that of the PLC-γ1 variants with the following modifications. After removal of the His6 tag by TEV protease and subsequent 5 mL HisTrap HP IMAC column, the sample was concentrated using a GE Healthcare VivaSpin 6 10K MWCO centrifugal concentrator. Concentrated sample was applied to a 16 mm × 700 mm HiLoad Superdex 200 size exclusion column equilibrated in buffer containing 20 mM HEPES (pH 7.5), 200 mM NaCl, 5% (v/v) glycerol, and 2 mM DTT. Fractions containing pure, kinase-inactive FGFR1K were concentrated to a final concentration of 20–30 mg/mL, aliquoted, snap-frozen in liquid nitrogen, and stored at –80°C until use. The kinase active form of FGFR1K harboring Y766F was expressed as described above; however, purification terminated after the first 5 mL HisTrap HP IMAC column so the protein retains its His6 tag.

## In vitro phosphorylation of FGFR1K
Equimolar concentrations (100 μM) of tagless, kinase-inactive FGFR1K and the tagged, kinase-active version were incubated in 20 mM HEPES (pH 7.5), 50 mM NaCl, 25 mM MgCl₂, 50 ng/mL fatty acid-free bovine serum albumin (FAF BSA), 10 mM ATP, and 2 mM DTT. The phosphorylation reaction was terminated after 100 min by adding EDTA to a final concentration of 50 mM and kinase phosphorylation was confirmed via native PAGE followed by staining with Coomassie Brilliant Blue. To separate the two forms of FGFR1K, the mixture was loaded onto a 1 mL HisTrap HP IMAC using a 200 μL sample loop followed by 2 mL of buffer N1 (20 mM HEPES pH 7.5, 100 mM NaCl, 10 mM MgCl₂, and 2 mM DTT). The column was washed with 5 CV of buffer N1, followed by 5 CV of 40% buffer N2 (buffer N1 + 1 M imidazole), and 5 CV of 100% buffer N2. Fractions containing phosphorylated, kinase-inactive FGFR1K were pooled and concentrated using a GE Healthcare Vivaspin 6 10K MWCO, aliquoted, snap-frozen in liquid nitrogen, and stored at –80°C until use.

## LC-MS/MS of phosphorylated FGFR1K
Phosphorylated, kinase-inactive FGFR1K was diluted and loaded onto native PAGE followed by staining with Coomassie Brilliant Blue. Gel bands were excised and digested with trypsin overnight. Peptides were extracted, then analyzed by LC-MS/MS using the Thermo Easy nLC 1200-QExactive HF. Data were searched against a UniProt Sf9 database including the sequence for FGFR1K using Sequest within Proteome Discoverer 2.1. All data were filtered using a false discovery rate of 5%.

### Formation of PLC-γ1 in complex with FGFR1K

Kinase-inactive FGFR1K phosphorylated at Tyr766 was generated as described above. Either PLC-γ1 (H335A) or PLC-γ1 (H335A+D1165H) was added in a two-fold molar excess relative to kinase. The sample was immediately loaded onto a 10 mm × 300 mm Superdex 200 GL size exclusion column equilibrated with 20 mM HEPES (pH 7.5), 100 mM NaCl, 10 mM MgCl$_2$, and 2 mM DTT. Pure complex was pooled and concentrated using a GE Healthcare Vivaspin 6 10K MWCO, aliquoted, snap-frozen in liquid nitrogen, and stored at –80°C until use. Formation of the complex was confirmed via SDS-PAGE followed by staining with Coomassie Brilliant Blue.

## In vitro quantification of phospholipase activity

### XY-69 fluorogenic assay

Liposomes with a final PE:PIP$_2$ content of 80:20 were generated by mixing 750 nM XY-69 (*Huang et al., 2018*), 192 µM of liver phosphatidylethanolamine (PE, Avanti Polar Lipids), 48 µM brain PIP$_2$ (Avanti Polar Lipids) in 12 × 75 mm borosilicate tubes. Lipids were dried under a nitrogen stream followed by high vacuum (0.5 mtorr). Dried lipid mixture was suspended in 20 mM HEPES (pH 7.5) using a probe microtip sonicator of 5/64" at 20% output for three cycles of 5 s ON, 15 s OFF. Concurrently, the PLC-γ1 proteins were diluted in a buffer containing 20 mM HEPES (pH 7.5), 50 mM NaCl, 2 mM DTT, and 1 mg/mL FAF BSA. The 6X assay buffer containing 80 mM HEPES (pH 7.5), 420 mM KCl, 10 mM DTT, 18 mM EGTA, and 14.1 mM CaCl$_2$ (~390 nM free Ca$^{2+}$) was added to the resuspended lipid mixture in a 1:4 ratio. To a non-binding surface-treated Corning 384-well plate, 2 µL of diluted PLC-γ1 proteins were added, either alone or in the presence of a two-fold molar excess of unphosphorylated or phosphorylated, kinase-inactive FGFR1K relative to PLC-γ1. To initiate the assay, 10 µL of the lipid and assay buffer mixture were added. The plates were incubated at 30°C and data was recorded for 30 min at intervals of 1 min using excitation and emission wavelengths of 485 nm and 520 nm, respectively. Fluorescence intensity was normalized to a blank reaction lacking phospholipase, and initial rates of XY-69 hydrolysis were calculated from the slope of the linear portion of the curve. Final concentrations of XY-69, PIP$_2$, and PE were 5 µM, 48 µM, and 192 µM, respectively.

### WH-15 fluorogenic assay

Assays with the soluble substrate WH-15 were performed as described previously (*Huang et al., 2011*) with the following modifications. WH-15 was diluted to a final concentration of 5 µM in assay buffer containing 50 mM HEPES (pH 7.5), 70 mM KCl, 3 mM EGTA, 2.97 mM CaCl$_2$ (~10 µM free Ca$^{2+}$), 50 µg/mL FAF BSA, and 2 mM DTT. Basal fluorescence was equilibrated for approximately 10 min before addition of PLC-γ1 (D1165H) alone or in the presence of a two-fold molar excess of unphosphorylated or phosphorylated, kinase-inactive FGFR1K relative to PLC-γ1 (1 nM, final concentration). Data were recorded for 1 hr at intervals of 30 s using excitation and emission wavelengths of 488 nm and 520 nm, respectively. Fluorescence intensity was normalized to the reaction lacking phospholipase, and initial rates of WH-15 hydrolysis were calculated from the slope of the linear portion of the curve.

### Liposome flotation assay

Liposomes with a final content of 89.8% PE, 10% PIP$_2$, and 0.2% 7-nitro-2–1,3-benzoxadiazol-4-yl-PE were dried down and resuspended in 20 mM HEPES (pH 7.5) by sonication as described above for assays using XY-69. Concurrently PLC-γ1 (H335A), alone or in complex with phosphorylated, kinase-inactive FGFR1K was diluted in assay buffer containing 100 mM HEPES (pH 7.5), 150 mM NaCl, 10 mM KCl, 10 µM CaCl$_2$, 0.5 mM tris(2-carboxyethyl)phosphine (TCEP). PLC-γ1 and liposomes were mixed and incubated on ice for 2 min. Immediately after mixing, 100 µL of assay buffer with 75% sucrose were added to the protein-lipid mixture to a final concentration of 30% sucrose in 250 µL total volume. The samples were overlaid with 200 µL of assay buffer with 25% sucrose and 50 µL of assay buffer. Samples were centrifuged at 55,000 rpm in a TLS-55 rotor for 1 hr at 4°C. After centrifugation, three fractions were collected: bottom (B, 250 µL), middle (M, 150 µL), and top (T, 100 µL). Samples (30 µL) of each fraction were mixed with 7 µL of 6X SDS sample buffer and analyzed by SDS-PAGE followed by staining with Coomassie Brilliant Blue.

## Hydrogen-deuterium exchange mass spectrometry

### Sample preparation

Exchange reactions were carried out at 18°C in 20 μL volumes with a final concentration of 1.25 μM PLC-γ1 (H335A), PLC-γ1 (H335A+D1165H), PLC-γ1 (H335A)-FGFR1K complex, or PLC-γ1 (H335A+D1165H)-FGFR1K complex. A total of eight conditions were assessed: four in the presence of liposomes containing 90% PE and 10% $PIP_2$ and four in the absence of liposomes. PLC-γ1 biochemical assays have showed similar activity of vesicles of varying composition (e.g., PC/PS/$PIP_2$/cholesterol), so the PE/$PIP_2$ mixture was used for its experimental simplicity. The conditions were as follows:

    PLC-γ1 (H335A) alone
    PLC-γ1 (H335A+D1165H) alone
    PLC-γ1 (H335A)−FGFR1K
    PLC-γ1 (H335A+D1165H)−FGFR1K
    PLC-γ1 (H335A)+liposomes
    PLC-γ1 (H335A+D1165H)+liposomes
    PLC-γ1 (H335A)−FGFR1K+liposomes
    PLC-γ1 (H335A+D1165H)−FGFR1K+liposomes

For conditions containing liposomes, lipids were present at a final concentration of 320 μM. Prior to the addition of $D_2O$, 2.0 μL of liposomes (or liposome buffer [20 mM HEPES pH 7.5, 100 mM KCl]) were added to 1.5 μL of protein, and the solution was left to incubate at 18°C for 2 min. Hydrogen-deuterium exchanges were initiated by the addition of 16.5 μL $D_2O$ buffer (88% $D_2O$, 150 mM NaCl, 100 mM HEPES pH 7.0, 10 μM $CaCl_2$, 0.5 mM TCEP pH 7.5)–3.5 μL protein or protein-liposome solution for a final $D_2O$ concentration of 72%. Exchange was carried out over five time points (3, 30, 300, 3000, and 10,000 s) and terminated by the addition of 50 μL ice-cold acidic quench buffer (0.8 M guanidine-HCl, 1.2% formic acid). After quenching, samples were immediately frozen in liquid nitrogen and stored at –80°C. All reactions were carried out in triplicate.

For PLC-γ1 (H335A) versus PLC-γ1 (H335A+D1165H), exchange reactions were carried out at 18°C in 50 μL volumes with a final concentration of 0.4 μM PLC-γ1 or PLC-γ1 (D1165H). Hydrogen-deuterium exchanges were initiated by the addition of 49 μL $D_2O$ buffer (89% $D_2O$, 100 mM NaCl, 50 mM HEPES pH 7.5)–1 μL protein for a final $D_2O$ concentration of 87%. Exchange was carried out over five time points (3 s on ice and 3, 30, 300, 3000 s at RT) and terminated by the addition of 20 μL ice-cold acidic quench buffer (2 M guanidine-HCl, 3% formic acid). After quenching, samples were immediately frozen in liquid nitrogen and stored at –80°C. All reactions were carried out in triplicate.

### Protein digestion and MS/MS data collection

Protein samples were rapidly thawed and injected onto an integrated fluidics system containing a HDX-3 PAL liquid handling robot and climate-controlled chromatography system (LEAP Technologies), a Dionex Ultimate 3000 UHPLC system, as well as an Impact HD QTOF Mass spectrometer (Bruker). Proteins were run over two immobilized pepsin columns (Applied Biosystems; Poroszyme Immobilized Pepsin Cartridge, 2.1 mm × 30 mm; Thermo-Fisher 2-3131-00; at 10 and 2°C, respectively) at 200 μL/min for 3 min. The resulting peptides were collected and desalted on a C18 trap column (Acquity UPLC BEH C18 1.7 μm column [2.1 × 5 mm]; Waters 186002350). The trap was subsequently eluted in line with a C18 reverse-phase separation column (Acquity 1.7 μm particle, 100 × 1 $mm^2$ C18 UPLC column, Waters 186002352), using a gradient of 5–36% B (buffer A 0.1% formic acid; buffer B 100% acetonitrile) over 16 min. Lipids were directly captured on the LC system, and eluted off at the 100% acetonitrile step, with no interference on mass spectrometer or LC performance. Full details of the LC setup and gradient for lipid samples are in *Stariha et al., 2021*. Mass spectrometry experiments were performed on an Impact II QTOF (Bruker) acquiring over a mass range from 150 to 2200 m/z using an electrospray ionization source operated at a temperature of 200°C and a spray voltage of 4.5 kV.

### Peptide identification

Peptides were identified using data-dependent acquisition following tandem MS/MS experiments (0.5 s precursor scan from 150 to 2000 m/z; twelve 0.25 s fragment scans from 150 to 2000 m/z). MS/MS datasets were analyzed using PEAKS7 (PEAKS), and a false discovery rate was set at 1% using a database of purified proteins and known contaminants (*Dobbs et al., 2020*).

## Mass analysis of peptide centroids and measurement of deuterium incorporation

HD-Examiner Software (Sierra Analytics) was used to automatically calculate the level of deuterium incorporation into each peptide. All peptides were manually inspected for correct charge state, correct retention time, absence of overlapping isotopic traces, and appropriate selection of isotopic distribution. Deuteration levels were calculated using the centroid of the experimental isotope clusters. Results for these proteins are presented as relative levels of deuterium incorporation and the only control for back exchange was the level of deuterium present in the buffer (72%). Changes in any peptide at any time point (3, 30, 300, 3000, and 10,000 s) greater than specified cut-offs (5% and 0.4 Da) and with an unpaired, two-tailed t-test value of p<0.01 were considered significant.

The raw peptide deuterium incorporation graphs for a selection of peptides with significant differences are shown (*Figure 3*, *Figure 2—figure supplement 4*, *Figure 4—figure supplement 1*), with the raw data for all analyzed peptides in the source Excel file. To allow for visualization of differences across all peptides, we used number of deuteron difference (#D) plots (*Figures 2–4*, *Figure 2—figure supplement 4*, *Figure 4—figure supplement 1*, *Figure 4—figure supplement 2*). These plots show the total difference in deuterium incorporation over the entire HDX time course, with each point indicating a single peptide. These graphs are calculated by summing the differences at every time point for each peptide and propagating the error (e.g. *Figure 2A*). For a selection of peptides, we are showing the %D incorporation over a time course, which allows for comparison of multiple conditions at the same time for a given region (*Figure 2—figure supplement 4*, *Figure 4—figure supplement 1*). Samples were only compared within a single experiment and were never compared to experiments completed at a different time with a different final D₂O level. The data analysis statistics for all HDX-MS experiments are in *Supplementary file 1* and conform to recommended guidelines (*Masson et al., 2019*). The MS proteomics data have been deposited to the ProteomeXchange Consortium via the PRIDE partner repository (*Perez-Riverol et al., 2019*) with the dataset identifier PXD030492.

## Acknowledgements

We acknowledge members of the Sondek and Burke labs for their valuable feedback and technical support. We also thank Laura Herring and Josh Beri at the UNC Michael Hooker Proteomics Center for assistance with LC-MS/MS of phosphorylated kinase. This work was supported by The National Institutes of Health Grants R01-GM057391 (JS) and R01-GM098894 (QZ and JS), the Natural Science and Engineering Research Council of Canada (JEB, Discovery grant NSERC-2020–04241) and the Michael Smith Foundation for Health Research (JEB, Scholar Award 17686). ES-P was supported by a National Science Foundation Graduate Research Fellowship under Grant No. DGE-1650116.

The content is solely the responsibility of the authors and does not necessarily represent the official views of the National Institutes of Health.

## Additional information

### Competing interests

John E Burke: Burke reports consulting fees from Scorpion Therapeutics and Olema Oncology, and research grants from Novartis, which are all outside the scope of this work. John Sondek: Partial ownership of KXTbio, Inc which licenses the production of WH-15. The other authors declare that no competing interests exist.

### Funding

| Funder | Grant reference number | Author |
| --- | --- | --- |
| National Institutes of Health | R01-GM057391 | John Sondek |
| National Institutes of Health | R01-GM098894 | Qisheng Zhang John Sondek |

| Funder | Grant reference number | Author |
|---|---|---|
| Government of Canada | Natural Science and Engineering Research Council of Canada NSERC-2020-04241 | John E Burke |
| Michael Smith Foundation for Health Research | Scholar Award 17686 | John E Burke |
| National Science Foundation | DGE-1650116 | Edhriz Siraliev-Perez |

The funders had no role in study design, data collection and interpretation, or the decision to submit the work for publication.

## Author contributions

Edhriz Siraliev-Perez, Conceptualization, Formal analysis, Investigation, Methodology, Visualization, Writing – original draft, Writing – review and editing; Jordan TB Stariha, Reece M Hoffmann, Meredith L Jenkins, Investigation, Methodology; Brenda RS Temple, Methodology; Qisheng Zhang, Methodology, Resources; Nicole Hajicek, Conceptualization, Formal analysis, Project administration, Supervision, Writing – review and editing; John E Burke, Conceptualization, Data curation, Formal analysis, Funding acquisition, Investigation, Methodology, Project administration, Software, Supervision, Validation, Visualization, Writing – review and editing; John Sondek, Conceptualization, Formal analysis, Funding acquisition, Project administration, Supervision, Writing – original draft, Writing – review and editing

## Author ORCIDs

Edhriz Siraliev-Perez ![ORCID] http://orcid.org/0000-0003-1824-863X
Brenda RS Temple ![ORCID] http://orcid.org/0000-0002-9233-0191
Nicole Hajicek ![ORCID] http://orcid.org/0000-0001-7457-4830
John E Burke ![ORCID] http://orcid.org/0000-0001-7904-9859
John Sondek ![ORCID] http://orcid.org/0000-0002-1127-8310

## Decision letter and Author response

Decision letter https://doi.org/10.7554/eLife.77809.sa1
Author response https://doi.org/10.7554/eLife.77809.sa2

# Additional files

## Supplementary files
• Transparent reporting form

• Supplementary file 1. Experimental parameters associated with hydrogen-deuterium exchange experiments.

## Data availability

All HDX-MS data have been deposited to the ProteomeXchange Consortium with dataset identifier PXC030492.

The following dataset was generated:

| Author(s) | Year | Dataset title | Dataset URL | Database and Identifier |
|---|---|---|---|---|
| Siraliev-Perez E, Stariha JTB, Hoffmann RM, Temple BRS, Zhang Q, Hajicek N, Jenkins ML, Burke JE, Sondek J | 2022 | Dynamics of allosteric regulation of the phospholipase C-gamma isoform | http://proteomecentral.proteomexchange.org/cgi/GetDataset?ID=PXD030492 | ProteomeXchange, PXD030492 |

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
