## [Editor Report]

This work provides insight into how phospholipase C-γ1 (PLC-γ1) becomes activated upon binding to phosphorylated receptor tyrosine kinase, with an analysis of PLC-γ1 bound to the soluble kinase domain of *FGFR1* (FGFR1K) and/or liposomes containing PIP_2_. The most interesting finding is that regions of the protein far from the FGFR1K binding site increase in exchange upon binding. This is new information for a large protein that is arguably difficult to study, but it conforms to what has been observed in many other autoinhibited systems with similar SH2 and SH3 domains such as kinases. The results will be of interest to structural biologists and cell biologists with interest in the mechanisms leading to the regulation of phospholipase C activity on membranes.

---

## [Decision Letter]

**Decision letter after peer review:**

Thank you for submitting your article "Dynamics of allosteric regulation of the phospholipase C-γ isozymes upon recruitment to membranes" for consideration by *eLife*. Your article has been reviewed by 2 peer reviewers, including Felix Campelo as Reviewing Editor and Reviewer #1, and the evaluation has been overseen by Jonathan Cooper as the Senior Editor.

Essential revisions:

1) Add additional controls/experiments/discussion:

– Clarify if the complex is completely bound under HDX conditions. (Rev. 2 #5)

– How does freezing affect the samples? (Rev. 2 #6)

– Use of different lipid compositions. (Rev. 1 #3)

– Compare results using phosphorylated vs. non-phosphorylated FGFR1K. (Rev. 1 #4)

2) Improve on the methodological description. (Rev. 2 #3,4)

3) Improve data visualization. (Rev. 1 #1; Rev. 2 #1)

4) Clarify some points in the text. (Rev. 1 #1; Rev. 2 #2)

*Reviewer #1 (Recommendations for the authors):*

I have some general questions and comments, which I hope might help the authors present a stronger/clearer paper:

1) Regarding the ability of WT vs H335A PLC to interact with PIP2-containing liposomes. From the results presented in Figure S1B, I interpret that WT binds poorly to PIP2 liposomes: T/M/B fractions in WT+liposomes (lanes 7-9) are apparently very similar to T/M/B fractions in WT-liposomes (lanes 1-4), whereas T/M/B fractions in H335+liposomes (lanes 10-12) show a stronger floated (T) fraction co-floating with lipids. The text seems to indicate something different (lines 129 on). Maybe I missed something, so a clarification of this point would be important.

2) Have the authors considered comparing HDX-MS experiments done in PLC-WT vs. H335A?

3) Regarding the role of PIP2, the authors only used a single liposome composition (90% PE + 10% PIP2). I understand these are long and difficult experiments, but why did the authors choose PE for liposomes and not something more "neutral" such as PC (PE is a lipid with a negative spontaneous curvature, which might give rise to the binding of proteins due to detection of lipid packing factors). Also, what about comparing results using liposomes with or without PIP2? Could the authors comment on this?

Related to this, have the authors tested the binding to liposomes of H335A vs. H335A+D1165H mutants?

4) To relate to the interesting findings presented in Figure 5B, could the authors compare their results using phosphorylated vs. non-phosphorylated FGFR1K?

*Reviewer #2 (Recommendations for the authors):*

There are several things that would be required for the manuscript to be acceptable:

1) Liposomes alone do not alter the structure of the protein, however when FGFR1K and liposomes are combined, the increases in exchange in the TIM barrel are more pronounced than in the presence of either ligand. Figure 2 shows the data but it is rather difficult to see differences between panel A and C. Perhaps including a difference plot between panel A and C would be helpful.

2) The authors occasionally use the word cooperativity for this dual action of liposomes and FGFR1K however they do not prove cooperativity as that requires demonstrating that binding of one ligand improves the binding of another ligand. The authors should refrain from the use of the word cooperativity in describing these results.

3) The authors then go on to study an oncogenic mutant which appears to only have significant changes in HDX at the site of mutation although at later times some small additional effects are seen. I couldn't find in the paper how the authors decided which time point(s) of HDX to use for their significance analysis and this should be explicitly stated. Similar changes are seen upon FGFR1K binding as were seen for the wild type protein but here, changes ARE seen upon binding liposomes alone mainly in the TIM barrel domain. These decreases then become increases upon binding both the FGFR1K and the liposomes.

4) The authors claim to identify nearly 300 peptides but given that they use a 16 min gradient and don't mention using ion mobility separation, it isn't clear how many of these peptides are analyzable without peptide overlap in the deuterated samples.

5) The binding affinities are all presumed, the manuscript does not contain any binding affinities and the SEC traces in Figure S3 clearly show separation between the PLCg1 and the FGFR1K when the complex is subjected to SEC so it isn't clear whether the complex is completely bound under the HDX conditions despite their using a 2-fold molar excess of FGFR1K.

6) It isn't clear how the samples are affected by freezing after deuteration and prior to pepsin cleavage since they contain liposomes. The authors should describe how the lipids separated from the proteins/peptides during HDX-MS and show controls for freezing before pepsin digestion vs immediate analysis of the deuterium exchanged samples.

---

## [Author Response]

Reviewer #1 (Recommendations for the authors):I have some general questions and comments, which I hope might help the authors present a stronger/clearer paper:1) Regarding the ability of WT vs H335A PLC to interact with PIP2-containing liposomes. From the results presented in Figure S1B, I interpret that WT binds poorly to PIP2 liposomes: T/M/B fractions in WT+liposomes (lanes 7-9) are apparently very similar to T/M/B fractions in WT-liposomes (lanes 1-4), whereas T/M/B fractions in H335+liposomes (lanes 10-12) show a stronger floated (T) fraction co-floating with lipids. The text seems to indicate something different (lines 129 on). Maybe I missed something, so a clarification of this point would be important.

The text has been clarified to read:

“In addition, a floatation assay was used to show that PLC-γ1 (H335A) retains capacity to associate with PIP_2_-containing liposomes.”

2) Have the authors considered comparing HDX-MS experiments done in PLC-WT vs. H335A?

This is a very good suggestion. We actually performed our pilot HDX-MS studies on wild-type PLC-γ1 with and without the oncogenic D1165H mutation. However, these experiments were performed under different exchange conditions to the H335A mutant experiments required for membrane binding (different % deuterium content, due to the presence of lipid) which prevents a simple direct comparison. We agree that this information could be valuable to include, as importantly, we saw almost exactly the same differences in exchanges caused by the oncogenic mutant in either the wild-type or H335A background. This strongly indicates that H335A has limited effect on the dynamic movements observed between the catalytic and regulatory domains.

We have added the raw deuterium incorporation information for this experiment into the source data, and added a brief description in the results and methods.

3) Regarding the role of PIP2, the authors only used a single liposome composition (90% PE + 10% PIP2). I understand these are long and difficult experiments, but why did the authors choose PE for liposomes and not something more "neutral" such as PC (PE is a lipid with a negative spontaneous curvature, which might give rise to the binding of proteins due to detection of lipid packing factors). Also, what about comparing results using liposomes with or without PIP2? Could the authors comment on this?

Excellent points; undoubtedly, the composition of the vesicles used for these studies do not fully recapitulate biological membranes. However, we have not observed substantial differences in the specific activity of PLC-γ1 when using vesicles of vary composition (e.g., PC/PS/PIP2/cholesterol), and the PE/PIP2 mixture was used for its experimental simplicity.

Related to this, have the authors tested the binding to liposomes of H335A vs. H335A+D1165H mutants?

An excellent suggestion, however, since we were principally concerned that PLC-γ1 harboring the active site mutation (H335A) retained binding to liposomes, this experiment was not done. However, we would expect H335A+D1165H to possibly have a higher affinity for liposomes since we expect D1165H to promote membrane engagement by destabilizing the autoinhibitory interface. Future work is planned to quantify affinities of a panel of oncogenic forms of PLC-γ1 and model liposomes.

4) To relate to the interesting findings presented in Figure 5B, could the authors compare their results using phosphorylated vs. non-phosphorylated FGFR1K?

Phosphorylated Tyr766 of *FGFR1* is required to recruit PLC-γ1 to membranes (Mohammadi M. et al., Nature 358, 681 (1992)). The Y766F mutant has been shown to be unable to associate with PLC-γ1, therefore, non-phosphorylated FGFR1K is not expected to bind appreciably to PLC-γ1, but this expectation was not tested explicitly.

Reviewer #2 (Recommendations for the authors):There are several things that would be required for the manuscript to be acceptable:1) Liposomes alone do not alter the structure of the protein, however when FGFR1K and liposomes are combined, the increases in exchange in the TIM barrel are more pronounced than in the presence of either ligand. Figure 2 shows the data but it is rather difficult to see differences between panel A and C. Perhaps including a difference plot between panel A and C would be helpful.

The difference plot in panel A of Figure 2 shows the difference in exchange between apo and pFGFR1K-bound PLC-γ1. Panel C now shows the combined effect between Apo PLC-γ1 and when PLC-γ1 is bound to pFGFR1K and liposomes. In a four condition HDX experiment like this, there are many additional difference plots that can be generated. We found these three to be the most informative to include in the main text figure.

What the reviewer seems to want to know is the difference that occurs stepwise, i.e., PLC-γ1 bound to pFGFR1K versus PLC-γ1 bound to both pFGFR1K and liposomes; or conversely, PLC-γ1 bound to liposomes versus PLC-γ1 bound to liposomes and pFGFR1K. These data are included in panels A+B of Figure S5. However, we agree that this point may have been confusing to find, and to make this information more accessible to readers, we have more clearly described these data in the legend of Figure 2 of the main text.

An identical situation hold for the combination of Figure 4 and Figure S6, and we have also added similar details to the figure legend for Figure 4.

2) The authors occasionally use the word cooperativity for this dual action of liposomes and FGFR1K however they do not prove cooperativity as that requires demonstrating that binding of one ligand improves the binding of another ligand. The authors should refrain from the use of the word cooperativity in describing these results.

We agree with the reviewer that a demonstration of cooperativity would require a direct testing that the binding of the first ligand increases binding to the 2^nd^ ligand. While this is likely the case, due to the complicated nature of protein membrane binding assays this experiment is outside the scope of this work, and we have qualified the text to mention “potential” or “functional” cooperativity where appropriate.

3) The authors then go on to study an oncogenic mutant which appears to only have significant changes in HDX at the site of mutation although at later times some small additional effects are seen. I couldn't find in the paper how the authors decided which time point(s) of HDX to use for their significance analysis and this should be explicitly stated. Similar changes are seen upon FGFR1K binding as were seen for the wild type protein but here, changes ARE seen upon binding liposomes alone mainly in the TIM barrel domain. These decreases then become increases upon binding both the FGFR1K and the liposomes.

The criteria for a change to be considered significant is as follows from the methods. Significant differences in exchange in any peptide required three specific conditions: greater than both a 5 % and 0.4 Da difference in exchange at any timepoint (3, 30, 300, 3000, and 10000 sec), and a two-tailed unpaired t-test of p<0.01. We have now added this information to the figure legend of Figures2, 3, and 4 as well.

The oncogenic mutant induced significant changes throughout multiple domains of the protein. The graph in Figure 3B shows significant peptides as red dots, with individual peptide incorporation graphs included in Panel C (significance indicated by asterisk)*.* Many of these regions do indeed show differences upon either pFGFR1K or liposome binding as well, which fits with our central hypothesis that pFGFR1K and liposome work together to disengage the regulatory domains from the core, and that this oncogenic mutation primes the system for disengagement, all leading to subsequent access of the catalytic core to lipid substrate.

4) The authors claim to identify nearly 300 peptides but given that they use a 16 min gradient and don't mention using ion mobility separation, it isn't clear how many of these peptides are analyzable without peptide overlap in the deuterated samples.

Our laboratory has extensive experience in using high resolution TOF MS to analyze complex protein mixtures (up to 600 kDa of unique sequence). To prevent complications from overlap we manually curate all HDX datasets in HDExaminer for removal of peptides that have overlapping isotope traces. We have added this detail to the methods.

5) The binding affinities are all presumed, the manuscript does not contain any binding affinities and the SEC traces in Figure S3 clearly show separation between the PLCg1 and the FGFR1K when the complex is subjected to SEC so it isn't clear whether the complex is completely bound under the HDX conditions despite their using a 2-fold molar excess of FGFR1K.

Agreed. While the ratio of PLC-γ1 and pFGFR1K isolated by size exclusion chromatography and SDS-PAGE is suggestive of a 1:1 complex in comparison to standards, this ratio was never quantified.

The main text has been changed to reflect this point:

“Consequently, the kinase domain of *FGFR1* was mutated to remove minor site of tyrosine phosphorylation to render the kinase resistant phosphorylation-dependent activation before phosphorylation of Tyr766 and formation of a stable complex with PLC-γ1 (H335A) (Figures S2-S3).”

However, the most important experiment validating high occupancy of the PLC-pFGR1K tested in HDX experiments is present in the gel filtration trace in Figure S3C. The blue trace shows the gel filtration trace with excess pFGFR1K, with two peaks, one for complex and one for excess pFGFR1K. When the isolate complex was rerun on gel filtration after 8 hrs on ice (red trace) the gel filtration shows only a single peak at the elution volume of the complex, and no free pFGFR1K. This is highly indicative of a stable complex of PLC-γ1 and pFGFR1K.

6) It isn't clear how the samples are affected by freezing after deuteration and prior to pepsin cleavage since they contain liposomes. The authors should describe how the lipids separated from the proteins/peptides during HDX-MS and show controls for freezing before pepsin digestion vs immediate analysis of the deuterium exchanged samples.

While we did not do an experiment directly testing the digestion efficiency of PLC with and without the freezing step, we have found using more than 20 different previously tested samples that we have enhanced denaturation and digestion when HDX samples were pre-frozen, versus set up at low temperature using our LEAP robotics setup. For this reason we now do all of our experiments with a pre-freezing step. As for lipid separation, this is all done on the LC fluidics setup, as if vesicles are extruded they can be run on the LC system with no complications. The full details of this are cited in methods from a methods chapter on using HDX-MS to study peripheral membrane proteins (PMID: 33877613). We have added details on this in the methods.